# Deep Networks Learn Features From Local Discontinuities in the Label Function

**Prithaj Banerjee, Harish G. Ramaswamy, Yadav Mahesh Lorik, Chandrashekar Lakshminarayanan**
Indian Institute of Technology Madras, India
`{{prithaj,maheshyadav}@cse, {hariguru,chandarashekar}@dsai}.iitm.ac.in`

## Abstract

Deep neural networks outperform kernel machines on several datasets due to feature learning that happens during gradient descent training. In this paper, we analyze the mechanism through which feature learning happens and use a notion of features that corresponds to discontinuities in the true label function. We hypothesize that the core feature learning mechanism is label function discontinuities attracting model function discontinuities during training. To test this hypothesis, we perform experiments on classification data where the true label function is given by an oblique decision tree. This setup allows easy enumeration of label function discontinuities, while still remaining intractable for static kernel/linear methods. We then design/construct a novel deep architecture called a Deep Linearly Gated Network (DLGN), whose discontinuities in the input space can be easily enumerated. In this setup, we provide supporting evidence demonstrating the movement of model function discontinuities towards the label function discontinuities during training. The easy enumerability of discontinuities in the DLGN also enables greater mechanistic interpretability. We demonstrate this by extracting the parameters of a high-accuracy decision tree from the parameters of a DLGN. We also show that the DLGN is competitive with ReLU networks and other tree-learning algorithms on several real-world tabular datasets.

## 1 Introduction

Neural networks and deep learning have demonstrated exceptional performance across diverse domains, but their learning mechanism and exact reason for outperforming linear/kernel models still remain largely a mystery. Another issue with neural networks is the black box nature of the models, by which any parameters (other than the first layer parameters) are difficult to interpret directly.

There have been several tools built to study the learning dynamics of gradient descent and perform post-hoc interpretation of learned models. The most common characteristics of these are as follows:

- Approximate the learning dynamics of neural networks with much simpler setups – like kernel methods (Arora et al., 2019) or deep linear models (Saxe et al., 2014).

- Study dynamics of single (or two) hidden layer models under special data and settings to go beyond the settings of kernel methods or deep linear models (Damian et al., 2022; Ba et al., 2022; Nichani et al., 2023).

- Approximate the learned non-linear neural model with linear approximations around a given query input (Ribeiro et al., 2016; Selvaraju et al., 2017).

These approaches serve a valuable purpose, but have their share of cons, namely

- Deep linear models are incapable of learning non-linear models, and fixed kernel methods are incapable of learning "data-dependent-features".

- Analyses of single hidden layer models for specific data settings like the parity function fail to capture the need/behaviour of multiple layers.

- All post-hoc interpretability approaches based on local approximations suffer from a lack of faithfulness guarantees– i.e. how useful/valuable is the 'interpreted' feature to the learned model?

To this end, we introduce and study a novel architecture which is more powerful than kernel machines and deep linear networks (i.e. it is capable of learning non-linear data-dependent features) while being 'mechanistically interpretable' (i.e. the model output breaks down exactly into a sum of components, each of which has a simple structure). Using this architecture, we gather and provide supporting evidence for claims below that would not be possible without it.

- The advantage of feature learning methods (like ReLU networks) over kernel machines lies in their ability to align the model discontinuities to the label function discontinuities.
- The existence of multiple layers enables the 'local' nature of the discontinuity finding – each layer operates in the context set by other layers, and hence, discontinuities that might be minor (or not even exist) in a global context can become significant and drive the learning process.

## 1.1 OVERVIEW/CONTRIBUTIONS

In Section 2, we review recent literature on feature learning in deep networks. In Section 3 we define the concept of label function discontinuities and give an interesting setting where these form natural features and can be easily enumerated. In Section 4 we propose a novel architecture called the deep linearly gated network (DLGN). In Section 5 we provide supporting evidence for our claim of label function discontinuities attracting model features during training. In Section 6 we demonstrate that the DLGN architecture enables better control and is more amenable to being mechanistically interpreted by constructing a decision tree directly from the parameters of learned DLGNs. In Section 7 we give empirical results on some binary classification datasets and show that feature learning methods (like the DLGN) outperform fixed kernel methods and are comparable to ensemble methods like random forests.

## 1.2 NOTATION/SETUP

We consider a binary classification task, with training set $S = \{(\mathbf{x}_1, y_1), \ldots, (\mathbf{x}_n, y_n)\}$ where $\mathbf{x}_i \in \mathbb{R}^d$ is drawn from some distribution $D$, and $y_i = f^*(\mathbf{x}_i) \in \{+1, -1\}$. The ultimate goal is to generalize well by finding a classifier $f : \mathbb{R}^d \rightarrow \{+1, -1\}$ such that $f(\mathbf{x}) = f^*(\mathbf{x})$ with high probability over $\mathbf{x}$ drawn from $D$. For any positive integer $a$ we denote the set $\{1, 2, \ldots, a\}$ as $[a]$. We denote by $\mathbf{1}$(condition) as a $\{0, 1\}$ valued variable that takes 1 if the condition is true and 0 otherwise.

## 2 RELATED WORK

A brief overview of some related work in the topic of neural network feature learning is given below.

**Neural Nets as Kernel Machines**: An infinite-width neural net at initialisation is equivalent to a Gaussian processes with a particular kernel, and learning can be viewed as computing the posterior of the label function, based on the training data likelihood (Lee et al., 2017). Jacot et al. (2018) introduced the Neural Tangent Kernel (NTK), which approximates an infinite-width neural network trained with gradient descent to a kernel machine. Arora et al. (2019) analysed the width at which neural networks act approximately similar to kernel machine, this number while being finite is much larger than typical widths seen in practical networks. Daniely & Malach (2020) showed that neural nets are not just equivalent to kernel-methods, and are capable of succeeding in situations where kernel methods provably fail. The constant NTK setting simply asserts no feature learning takes place, in contrast to empirical results. Tools to study the change of the NTK features during training are an interesting and active area of research. (Atanasov et al., 2022; Damian et al., 2022; Hu et al., 2020; Ba et al., 2022; Chizat et al., 2019; Liu et al., 2020).

**Feature Learning in Neural Nets**: ReLU networks (Fukushima, 1969) have been a workhorse of deep learning and are the current focus of several theoretical results that aim to explain the success of

deep learning over kernel methods (Ghorbani et al., 2020). ReLU networks can learn important data characteristics known as "features", though the term can be ambiguous. Neural networks undergo feature learning, developing data representations during the process (Damian et al., 2022). Some research proposes that neural nets learn low-frequency functions (Rahaman et al., 2019; Cao et al., 2019) and simple features (Shah et al., 2020) first while training, and as the iteration progresses, it gradually learns more complex features (Kalimeris et al., 2019; Hu et al., 2020). Other schools of thought on this being the last layer neurons (Daniely, 2017; Lee et al., 2019) are a natural feature choice as the prediction function can simply be viewed as a linear function of the last layer.

**Modifying neural nets to understand feature learning**: A ReLU network can be viewed as a deep linear network with an input-dependent sub-network selection. Lakshminarayanan & Vikram Singh (2020) make this explicit by constructing an alternate architecture which builds two separate models – one for input-dependent sub-network selection and another for the deep linear network that the input goes through – and showed that feature learning happens via learning of active sub-networks . Disjunctive normal networks (Sajjadi et al., 2016) that utilize the disjunctive normal form, Gated Linear Networks (GLN) (Veness et al., 2021; Budden et al., 2020; Saxe et al., 2022) a backpropagation-free architecture where each unit (neuron) learns locally using data-dependent gating (context) are some other recent papers that enable a better study of feature learning in deep networks by constructing a different and (arguably) simpler architecture than standard ReLU nets.

## 3    LOCAL DISCONTINUITIES IN THE LABEL FUNCTION

From the perspective of a learning algorithm trying to fit a classification model on the training data, the only visible aspect is a $d$-dimensional scatter plot with labels coloured by their label. From this viewpoint, the most interesting aspects are regions of the input space that see a colour change. This has been well studied for linearly separable data using the language of geometry and maximum-margins. Extending this idea to linearly-non-separable data is tricky. We try to capture this behaviour using a notion of 'discontinuities' in the label function. Intuitively, the number of such discontinuities has to be small for the problem to be learnable with a reasonable number of data points (Klivans et al., 2008).

We define the local discontinuity coefficient (LDC) of a contiguous region $\mathcal{R} \subseteq \mathbb{R}^d$ and a manifold $\mathcal{M} = \{\mathbf{x} \in \mathbb{R}^d : f(\mathbf{x}) = 0\}$ cutting through the region $\mathcal{R}$ for a function $f : \mathbb{R}^d \to \mathbb{R}$ as follows:

$$\gamma(\mathcal{R}, f) = \tfrac{1}{2}\left|\mathbb{E}\Big[Y \mid X \in \mathcal{R}, f(X) > 0\Big] - \mathbb{E}\Big[Y \mid X \in \mathcal{R}, f(X) < 0\Big]\right|$$

where the expectation is w.r.t $(X, Y)$ with $X \sim D$ and $Y = f^*(X)$. We will call $\gamma(\mathbb{R}^d, f)$ as the global discontinuity coefficient (GDC) of $f$. Note that the LDC is a property of the data distribution $D$ and labelling function $f^*$. We will say that the scope of discontinuity at $f$ is $\mathcal{R}$ if $\gamma(\mathcal{R}, f)$ is large.

For example, if the data can be separated by a hyperplane $\{x : \mathbf{w}^\top \mathbf{x} + b = 0\}$, then $\gamma(\mathbb{R}^d, \mathbf{w}^\top \mathbf{x} + b) = 1$, and the scope of discontinuity of $\mathbf{w}^\top \mathbf{x} + b$ is the entire space $\mathbb{R}^d$. The region $\mathcal{R}$ corresponds to the scope of the discontinuity, and $\gamma(\mathcal{R}, f)$ captures the accuracy of the classifier $\mathrm{sign}(f(x))$ within this scope. We argue that such local discontinuities in the label function are what drive feature learning in deep networks.

As an illustrative running example, we consider a labelling function $f^*$ given by an Oblique Decision Tree (ODT) (Bertsimas & Dunn, 2017; Murthy et al.; 1994; Wickramarachchi et al., 2016; Carreira-Perpinán & Tavallali, 2018). ODTs are decision trees with linear internal node functions $\mathbf{u}^\top \mathbf{x} + b$, and a data point branches left/right depending on the sign of this value. These generalise standard decision trees. ODT labelling functions are particularly hard to learn for fixed-feature learning methods such as kernel machines, but allows for easy enumeration of all its discontinuities

In particular, we consider complete ODTs with node hyperplanes orthogonal to each other and each leaf node having an equal number of data points (we call these COB-ODTs short for complete orthogonal balanced ODTs). COB-ODTs are particularly hard for classic greedy tree learning methods such as CART (Breiman et al., 1984) to learn. This is because the hyperplanes corresponding to internal nodes (and the root node in particular) are all balanced (they have an equal number of positive and negative data points on both sides of the hyperplane). Hence they do not get picked by greedy

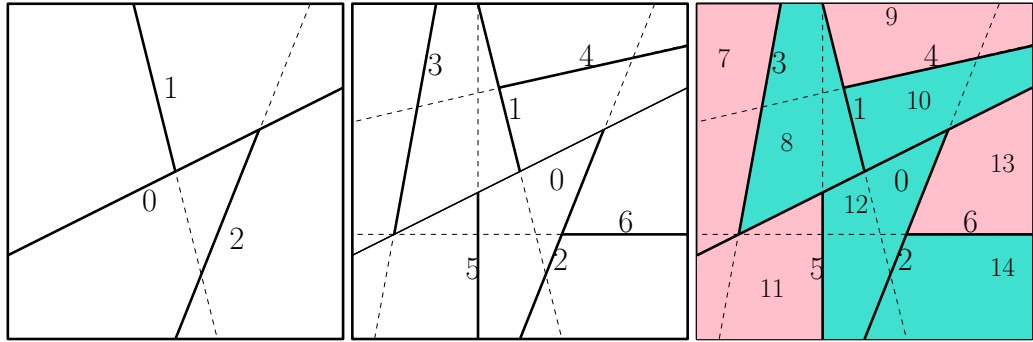

(a) ODT h-planes of root, children  (b) ODT h-planes of grandchildren  (c) Shaded leaf node regions.

Figure 1: Hyperplanes and labelling function for a complete ODT of depth 3 over a 2d input space. The children of an internal node $i$ are $2i+1$ and $2i+2$. Green and red shades indicate positive and negative labels respectively. Note that the hyperplanes are not orthogonal here – the input dimension must be greater than the number of internal nodes for a COB ODT to exist.

methods under metrics such as information gain or Gini-Index. Other ODT learning methods that are purportedly global (Zantedeschi et al., 2020; Bertsimas & Dunn, 2017; Lee & Jaakkola, 2020) also fail to identify internal node hyperplanes for unknown reasons. Finite decision trees that do not contain the internal node hyperplanes in a COB-ODT label function are guaranteed to have poor test error. On several such COB-ODT settings, where all fixed-feature learning methods and tree algorithms fail, deep ReLU nets and another deep architecture introduced in the next section succeed. Experimental results supporting these observations are given in Table 2.

It is important to note that the entire hyperplane corresponding to an internal node is not a discontinuity. For example, in the ODT in Figure 1(c), only parts of the line corresponding to root node 0 (or internal nodes 1 to 6) is a discontinuity. Also note that, despite the importance of the hyperplane corresponding to node 0, it would not be picked first by any greedy tree learner – the ratio between a number of positive and negatively labelled points is approximately the same on both sides of the hyperplane. This is exactly true for any data labelled by a COB-ODT.

Clearly, all the internal node hyperplanes all correspond to label function discontinuities – i.e. a randomly chosen point close to the hyperplane boundary has a chance of flipping labels if it moves to the other side of the hyperplane. No other hyperplane has this property. All internal node linear functions $f$ have an appropriate region $\mathcal{R}$ such that $\gamma(\mathcal{R}, f) = 1$, but the total mass of the region $\mathcal{R}$, given by $P(X \in \mathcal{R})$ is of the order of $2^{-\text{depth(internal node)}}$. An illustration of the computation of $\gamma$ is given in the Appendix.

To test our hypothesis of 'model features align with data features' we need a deep network architecture whose features (which are also simply discontinuities in the model function) can be easily enumerated. To this end we propose the deep linearly gated network architecture.

## 4    DEEP LINEARLY GATED NETWORKS(DLGN)

The DLGN architecture serves as a more transparent stand-in for ReLU networks and can be viewed as "a midway point between deep linear networks and ReLU networks". DLGNs (like ReLU nets) learn useful non-linear features and outperform kernel methods while having structural simplicity (like deep linear networks).

### 4.1    ARCHITECTURE DETAILS

The deep linearly gated network has an architecture similar to a ReLU network and is defined by neurons residing in multiple layers. For simplicity, we assume the architecture consists of $L$ hidden layers with $m$ neurons in each layer. The architecture is parameterized by matrices $W^1, W^2, \ldots, W^L$ and $U^2, \ldots, U^L$ and vectors $\mathbf{u}^1, \mathbf{u}^{L+1}$. The matrices $W^2, \ldots, W^L$ and $U^2, \ldots U^L$ are all of shape $m \times m$. $W^1$ has shape $m \times d$. $\mathbf{u}^1$ and $\mathbf{u}^{L+1}$ are vectors of size $m$.

The architecture is most naturally described using the notion of paths, which we denote by $\pi = (i_1, \ldots, i_L) \in [m]^L$, giving the sequence of hidden nodes that the path consists of. Let $\Pi = [m]^L$ denote the set of all paths in the network. The output of the model is given as follows:

$$\widehat{y}(\mathbf{x}) = \sum_{\pi \in \Pi} g_\pi f_\pi(\mathbf{x}) \tag{1}$$

where $f_\pi$ is called the path gating function for path $\pi$ and $g_\pi \in \mathbb{R}$ is the value of path $\pi$. The path gating function $f_\pi$ is defined by (what we call) the gating network – a deep linear network with weights $W^1, \ldots, W^L$. The path gating function $f_\pi$ is decomposed as a product of individual neuron gating functions that make up the path $\pi$. The path gating function for path $\pi = (i_1, \ldots, i_L)$ is

$$f_\pi(\mathbf{x}) = \prod_{\ell=1}^{L} \mathbf{1}\left(\boldsymbol{\eta}_{i_\ell}^\ell(\mathbf{x}) \geq 0\right) \tag{2}$$

$$\forall \ell \in [L], \qquad \boldsymbol{\eta}^\ell(\mathbf{x}) = W^\ell \boldsymbol{\eta}^{\ell-1}(\mathbf{x}) = V^\ell \mathbf{x} \tag{3}$$

where $\boldsymbol{\eta}^0(\mathbf{x}) = \mathbf{x}$ and $\forall \ell \in [L]$ the matrices $V^\ell \in \mathbf{R}^{m \times d}$ form the 'effective' weights of the neurons in layer $\ell$ and are given as $V^\ell = W^\ell W^{\ell-1} \ldots W^1$. We call the hyperplane $\{\mathbf{x} : \boldsymbol{\eta}_{i_\ell}^\ell(\mathbf{x}) = 0\}$ as the gating hyperplane corresponding to neuron $i_\ell$ in layer $\ell$.

The value $g_\pi$ of a path $\pi = (i_1, \ldots, i_L)$ is also defined by a network called the value network – a deep linear network with weights $U^2, \ldots, U^L, \mathbf{u}^{L+1}$, no biases, and input given by $\mathbf{u}^1$. It is simply the product of weights along the path $\pi$, i.e.

$$g_\pi = \mathbf{u}_{i_1}^1 \left[\prod_{\ell=2}^{L} U_{i_\ell, i_{\ell-1}}^\ell\right] \mathbf{u}_{i_L}^{L+1} \tag{4}$$

The model as defined in Equation (1) seems computationally hard to implement in a forward pass, but due to standard matrix multiplication properties, can be easily implemented at a cost that is less than twice the cost of a ReLU net with the same $mL$ hidden nodes. i.e.

$$\widehat{y}(\mathbf{x}) = \langle \mathbf{u}^{L+1}, h^L(\mathbf{x}) \rangle \tag{5}$$

where $h^1(\mathbf{x}) = \mathbf{1}(\boldsymbol{\eta}^1(\mathbf{x}) \geq 0) \circ \mathbf{u}^1$ and $h^\ell(\mathbf{x}) = \mathbf{1}(\boldsymbol{\eta}^\ell(\mathbf{x}) \geq 0) \circ \left(U^\ell h^{\ell-1}(\mathbf{x})\right)$ for $\ell > 1$. The gates $\boldsymbol{\eta}$ are as defined in Equation 3. The symbol $\circ$ represents elementwise multiplication. A proof for this equality is given in the Appendix along with a short note on how the DLGN is related to the ReLU network.

In order to learn the gating function parameters and back-propagate the gradient to $W$, we replace the indicator function by a sigmoid. i.e. $\mathbf{1}(a \geq 0)$ with $\sigma(\beta a)$ where $\sigma$ is the standard sigmoid function and $\beta > 0$ is a hyperparameter. A figure illustrating the DLGN architecture with an example is given in the Appendix.

## 4.2 VARIANTS

We also consider two natural variants of the DLGN architecture. The first is a simple re-parameterisation, where the gating network $f_\pi$ is parameterised directly by the matrices $V^1, V^2, \ldots, V^L$, each of shape $m \times d$, instead of using the parameters $W^1, \ldots, W^L$. Clearly, this reparameterisation does not lose any representation power. We call this variant as DLGN-SF (for shallow features).

In another variant, we use an explicit parameterisation of the coefficients $g_\pi$ instead of using a value network. This parameterisation is clearly more powerful than the standard DLGN parameterisation in the previous section. However, this requires a parameter tensor of size $m^L$ and is not practical for large $L$ and $m$. We call this variant as DLGN-VT (for value tensor).

## 4.3 DISCONTINUITY ENUMERATION

The model $\widehat{y}$ in Equation equation 1 is a linear combination of path gating functions. The discontinuities in $\widehat{y}$ are simply the union of discontinuities of $f_\pi$ over all paths $\pi$. While there are exponentially many gating functions $f_\pi$, they decompose further into a product of neuron gating functions, which are indicator functions over half-spaces. Thus, the set of all discontinuities can very simply be enumerated by considering the $mL$ hyperplanes corresponding to the neuron gating functions.

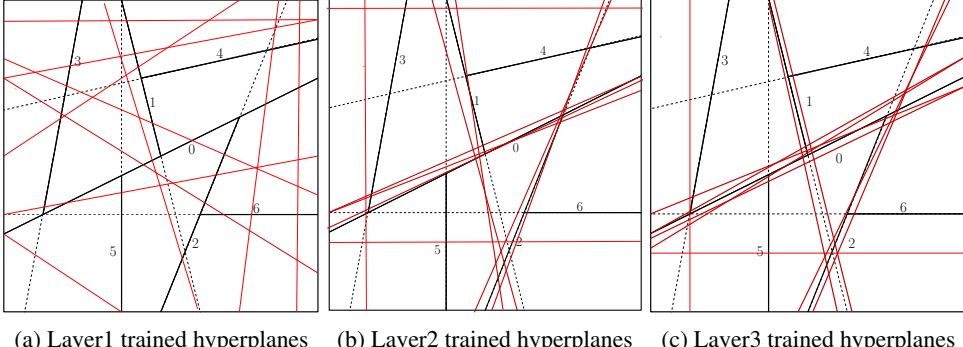

(a) Layer1 trained hyperplanes    (b) Layer2 trained hyperplanes    (c) Layer3 trained hyperplanes

Figure 2: An illustration of DLGN hyperplanes after training on data in Fig 1c.

## 5 TRAINING ATTRACTS MODEL FEATURES TOWARDS LABEL FEATURES

We now have a labelling function $f^*$ based on a COB-ODT and a model function $\widehat{y}$ based on a DLGN. The discontinuities of both $f^*$ and $\widehat{y}$ can be easily enumerated, and the discontinuities of both functions are in the form of hyperplanes. We can now evaluate the main claim of the paper.

A DLGN trained on data labelled by a COB-ODT shows several interesting properties. The most important of these properties is a tendency of the effective hyperplane of the gating neurons given by $V_i^\ell$ to cluster around the ODT node hyperplanes.

Figure 2 illustrates an example scenario when a 3-hidden layer DLGN is trained on data given in Figure 1(c). The initial hyperplanes (lines in this case) $\boldsymbol{\eta}^\ell(\mathbf{x}) = \mathbf{0}$ as shown in Appendix Figure 5(a-c) are essentially random. However, after training, the hyperplanes in the later layers show a remarkable tendency to move towards the hyperplanes corresponding to the decision tree – particularly that of nodes close to the root (See Figures 2(a-c)). The relative idleness of the first layer hyperplanes seems to be an artefact of parameterising the gating network by a deep linear network. This does not happen for DLGN-SF where the gating network effective weights $V^\ell$ are parameterised directly.

Table 1: Number of DLGN hyperplanes (after training) within a given distance of the label function ODT hyperplanes. The 15 ODT internal nodes are numbered 0 to 14, with 0 as the root. The distance of the closest (initial and trained) DLGN hyperplane to all the ODT hyperplanes is given in the last 2 rows. The DLGN has 4 hidden layers with 20 neurons in each layer, i.e. $L = 4, m = 20$.

| Distance | 0 | 1 | 2 | 3 | 4 | 5 | 6 | 7 | 8 | 9 | 10 | 11 | 12 | 13 | 14 |
|---|---|---|---|---|---|---|---|---|---|---|---|---|---|---|---|
| 0.1 | 4 | 2 | 2 | 1 | 1 | 1 | 1 | 1 | 1 | 1 | 1 | 1 | 1 | 1 | 1 |
| 0.2 | 12 | 3 | 5 | 2 | 1 | 1 | 1 | 1 | 1 | 1 | 1 | 1 | 1 | 1 | 1 |
| 0.3 | 13 | 3 | 5 | 2 | 1 | 1 | 1 | 2 | 1 | 1 | 1 | 1 | 1 | 2 | 1 |
| Closest Dist.(init) | 1.26 | 1.22 | 1.21 | 1.25 | 1.23 | 1.24 | 1.23 | 1.17 | 1.27 | 1.28 | 1.17 | 1.23 | 1.27 | 1.22 | 1.26 |
| Closest Dist.(final) | 0.05 | 0.07 | 0.06 | 0.07 | 0.1 | 0.06 | 0.05 | 0.06 | 0.09 | 0.08 | 0.07 | 0.07 | 0.06 | 0.09 | 0.06 |

Note that Figure 2 is a schematic illustration, and the decision tree hyperplane-seeking behaviour of the DLGN hyperplanes is exaggerated to illustrate the idea properly. Table 1 gives the results of a typical experiment on a synthetic dataset in which the data is a 100-dimensional vector uniformly distributed on the surface of the unit sphere and labelled by a depth-4 COB-ODT with 15 internal nodes. A 4-hidden layer DLGN with 20 neurons in each layer was trained on this dataset containing 30000 data points. Table 4 gives the same for a DLGN architecture with 100 neurons in each hidden layer. For each node in the ODT, we count the number of DLGN hyperplanes within a distance of $0.1, 0.2$, and $0.3$ from it. The distance between two hyperplanes $H(\mathbf{v}) = \{\mathbf{x} : \mathbf{v}^\top \mathbf{x} = 0\}$ and $H(\mathbf{z}) = \{\mathbf{x} : \mathbf{z}^\top \mathbf{x} = 0\}$ is $\min \left( \left\| \frac{\mathbf{z}}{\|\mathbf{z}\|} - \frac{\mathbf{v}}{\|\mathbf{v}\|} \right\|, \left\| \frac{\mathbf{z}}{\|\mathbf{z}\|} + \frac{\mathbf{v}}{\|\mathbf{v}\|} \right\| \right)$.

For large $d$, most pairs of vectors are orthogonal to each other, and hence, a typical value of $\text{dist}(\mathbf{v}, \mathbf{z})$ is about $\sqrt{2}$. At initialisation, there are almost no DLGN hyperplanes close (say distance less than 0.3) to any of the ODT hyperplanes. But it can be clearly seen from Table 1 that the training process attracts the DLGN hyperplanes towards the ODT hyperplanes. This happens most notably for the

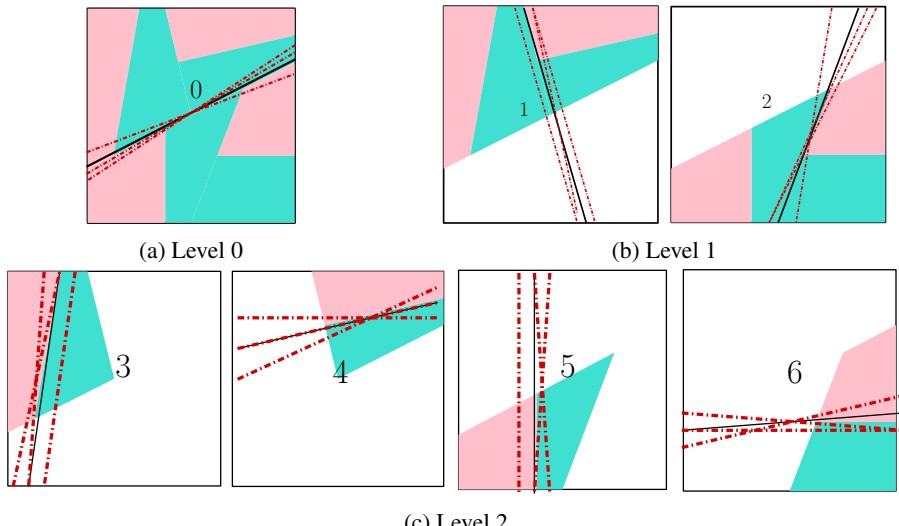

(a) Level 0

(b) Level 1

(c) Level 2

Figure 3: DLGN based ODT Learning. Illustration of the recursive procedure where for each level starting at 0, we have $2^{\text{level}}$ DLGNs trained on different splits of the data given by previous levels. The main result of each training run is the largest cluster of the learned DLGN hyperplanes, which is shown in the figures by red dashed lines.

root node, (13 out of the 80 DLGN gating hyperplanes end up close to the root node hyperplane), despite the root node having almost zero information gain or accuracy increase in the greedy decision tree construction setting. We conjecture that discontinuities in the label function serve as attractors for the DLGN gating hyperplanes under SGD. The strength of this attraction towards a hyperplane $f(\mathbf{x}) = 0$ is proportional to the scope of the label function discontinuity around this hyperplane (or the largest region $\mathcal{R}$ for which $\gamma(\mathcal{R}, f) = 1$).

We believe that there is a fundamental principle of feature learning in deep networks at play here, and studying the reason for this behaviour and proving it theoretically is a promising future direction that is beyond the scope of this paper.

## 6 MECHANISTIC INTERPRETABILITY OF DLGNS

The exact decomposition of the DLGN as a sum of the product of half-space indicator functions allows for a degree of control and reusability of the learned parameters that are not possible with a black box architecture like the ReLU network. We demonstrate this aspect of DLGNs by constructing a completely different type of model, namely an ODT, directly by copying parameters from learned DLGNs.

The decision tree extraction procedure is based on the principle conjectured in the previous section. Figure 3 gives a schematic illustration of the procedure. In the first stage, a DLGN is trained on the entire data. Our conjectured principle would then imply that the hyperplane with the largest scope of discontinuity (which would be the hyperplane of the root node of the ODT corresponding to $f^*$) would attract more DLGN hyperplanes than others, resulting in a detectable cluster around it. A clustering is performed over the learned DLGN hyperplanes, and the largest cluster is chosen (See Figure 3(a)). The cluster centre corresponding to it is chosen as the root node in the final decision tree. Based on the root node hyperplane, the training data can be split into two halves, and the procedure can be repeated on both halves recursively (See Figure 3(b,c)) until the data for training becomes too small or contains only one class. An oblique decision tree can thus be constructed by incorporating the largest cluster centres of the trained DLGN hyperplanes in the appropriate nodes of a tree.

The details of the above procedure are given in Algorithm 1. It returns a decision tree consisting of internal nodes and leaf nodes. Internal nodes are represented by a hyperplane and pointers to two child nodes. Leaf nodes are represented by a value that is either $+1$ or $-1$. The key subroutine in

---

**Algorithm 1** Building a decision tree from trained DLGN

---

**Arguments:** Binary classification training set containing pairs $(\mathbf{x}_i, y_i)$
**Outputs:** Root of an ODT

1: **function** BUILDTREE(data)
2:     **if** IMPURE(data) **and** LARGE(data) **then**
3:         left,right,$\mathbf{v}^* \leftarrow$ DISCONTHYPERPLANE(data)
4:         leftST $\leftarrow$ BUILDTREE(left)
5:         rightST $\leftarrow$ BUILDTREE(right)
6:         **return** NODE($\mathbf{v}^*$, leftST, rightST)
7:     **end if**
8:     lv $\leftarrow$ MAJORITYLABEL(data)
9:     **return** NODE(*value*=**lv**)
10: **end function**

---

**Algorithm 2** Finding Discontinuous Hyperplane

---

**Arguments:** Binary classification training set containing pairs $(\mathbf{x}_i, y_i)$
**Outputs:** Data split into 2 halves along a hyperplane, and the hyperplane parameters $\mathbf{w}$

1: **function** DISCONTHYPERPLANE(data)
2:     model $\leftarrow$ TRAINDLGN(data)
3:     $V \leftarrow$ GATEHYPERPLANES(model)
4:     $\mathbf{v}^* \leftarrow$ LARGESTCLUSTERCENTER($V$)
5:     left, right $\leftarrow$ SPLITDATA(data,$\mathbf{v}^*$)
6:     **return** left, right,$\mathbf{v}^*$
7: **end function**

---

Algorithm 1 is the DISCONTHYPERPLANE function detailed in Algorithm 2. It trains a DLGN on a classification dataset, clusters the DLGN hyperplanes, and splits the data based on this hyperplane. In our experiments, we used the DBScan (Ester et al., 1996) algorithm for clustering the DLGN hyperplanes as it is robust to outliers.

We call this end-to-end procedure of constructing an ODT from learned DLGNs as DLGN-DT. The total complexity of the DLGN-DT procedure is about depth times the cost of learning a single DLGN – even though $2^{\text{depth}}$ DLGNs are learned, most of them are learned on significantly smaller subsets of the data, and the complexity of clustering simply depends on the architecture size which is typically a small constant.

We emphasise that we are not proposing DLGN-DT as an alternate decision tree procedure. The main purpose of the DLGN-DT procedure is to show that we have better control and understanding of parameters in the DLGN. This is analogous to a mechanic demonstrating an understanding of a car, by dismantling it and building two motorcycles and a diesel electricity generator from it. This is fundamentally different from reusing the first few layers of a pre-trained convolutional net for feature extraction for the following reasons. Firstly, the atomic element that is copied is at the level of parameters and not sub-networks. Secondly, the new model is a decision tree which is a completely different type of model from neural networks trained via gradient descent. Thirdly, the new architecture (ODT) is not fine-tuned or retrained based on the data.

## 7 EXPERIMENTS

DLGN and its variants are more interpretable than a ReLU architecture, while still having powerful feature learning capabilities. Thus, they form an important tool in understanding the learning process of deep neural network. DLGNs in the special setting of a COB-ODT label function can also be used to illustrate the core feature learning hypothesis in the paper. In this section, we experimentally show that DLGNs, apart from being more interpretable, outperform static kernel methods, tree and non-tree algorithms and ReLU methods on COB-ODT synthetic datasets and are comparable in performance to ensemble tree methods on real tabular datasets.

Table 2: Test accuracy on synthetic datasets

| Dataset char. | | | DLGN | | | Non tree algo | | | | | | Tree algo | | | | |
|---|---|---|---|---|---|---|---|---|---|---|---|---|---|---|---|---|
| Data | n | d | **dlgn** | **dlgns** | **dlgnv** | relu | disnn | gln | svl | svm | sdt | cart | rf | zan | tao | **dlgnd** |
| SDI | 20000 | 20 | 96.1 | 97.4 | **98.4** | 90.6 | 89.8 | 66.8 | 63.4 | 76.5 | 68.9 | 57.3 | 65.9 | 84.9 | 60.3 | 93.1 |
| SDII | 30000 | 100 | 94.3 | 90.3 | **95.8** | 82.3 | 74.8 | 61.3 | 61.9 | 66.8 | 62.2 | 54.5 | 58.8 | 64.7 | 56.1 | 88.2 |
| SDIII | 50000 | 500 | **65.1** | 63.6 | 62.2 | 62.1 | 61.6 | 59.6 | 60.7 | 62.3 | 61.9 | 51.5 | 54.6 | 61.3 | 52.7 | 62.5 |

Table 3: Test accuracy on tabular datasets

| Dataset char. | | | DLGN | | Non tree algo | | | | | | Tree algo | | | |
|---|---|---|---|---|---|---|---|---|---|---|---|---|---|---|
| Data | n | d | **dlgn** | **dlgnv** | relu | disnn | gln | svl | svm | sdt | cart | rf | zan | tao |
| Adult | 29000 | 14 | 85.7 | 85.0 | 83.2 | 84.6 | 83.3 | 81.1 | 84.2 | 84.8 | 83.0 | **86.1** | 84.5 | 85.1 |
| Bank | 27000 | 16 | **91.6** | 91.3 | 89.6 | 91.2 | 90.4 | 89.9 | 90.7 | 91.3 | 90.3 | 91.3 | 91.1 | 90.9 |
| Card | 18000 | 23 | **81.6** | 81.2 | 78.4 | 80.9 | 80.4 | 80.6 | 81.1 | 81.3 | 77.6 | 81.2 | 81.3 | 80.6 |
| Telesc | 11000 | 10 | **88.2** | 87.2 | 87.7 | 87.7 | 85.6 | 79.4 | 87.1 | 86.4 | 82.9 | 88.1 | 87.0 | 84.9 |
| Rice | 2000 | 7 | **92.6** | 92.6 | 92.3 | 92.5 | 92.3 | 91.2 | 91.4 | 92.6 | 91.9 | 91.8 | 92.2 | 90.1 |
| Stat | 100 | 20 | **77.6** | 72.6 | 72.2 | 76.5 | 74.0 | 71.1 | 71.7 | 75.2 | 65.7 | 77.4 | 75.3 | 68.0 |
| Spam | 3000 | 57 | 94.4 | 94.0 | 94.1 | 94.0 | 92.1 | 92.3 | 93.1 | 93.4 | 89.4 | **94.8** | 93.1 | 91.2 |
| Gyro | 19000 | 8 | 98.8 | 98.3 | 98.4 | 98.4 | 98.3 | 98.3 | 98.1 | 98.4 | 98.7 | **99.1** | 98.6 | 98.6 |
| Swar | 14000 | 2400 | **100** | **100** | **100** | 87.5 | 99.6 | **100** | **100** | **100** | 99.9 | **100** | **100** | 99.9 |
| Credit | 12000 | 10 | 76.2 | 75.4 | 75.5 | 75.9 | 70.2 | 70.4 | 74.5 | 74.6 | 76.1 | **78.2** | 75.4 | 76.2 |
| Elec | 27000 | 7 | 82.8 | 80.6 | 82.7 | 79.6 | 77.5 | 73.6 | 78.5 | 75.8 | 86.3 | **87.6** | 76.5 | 76.4 |
| Cover | 396000 | 10 | **94.7** | 93.3 | 92.9 | 81.1 | 78.0 | 78.3 | 80.4 | 77.4 | 91.6 | 94.2 | 78.6 | 84.9 |
| Pol | 7000 | 26 | **98.9** | 98.0 | 98.2 | 97.9 | 93.8 | 87.8 | 97.7 | 98.2 | 96.6 | 97.4 | 97.6 | 95.4 |
| House | 9000 | 16 | 87.8 | 86.7 | 87.4 | 86.5 | 85.6 | 83.0 | 87.2 | 86.3 | 84.0 | **88.0** | 85.3 | 85.1 |
| Mini | 51000 | 50 | **93.6** | 93.4 | 93.2 | 91.0 | 88.6 | 89.3 | 89.3 | 88.9 | 89.9 | 93.1 | 91.7 | 88.8 |
| Diab | 50000 | 7 | **60.6** | 60.6 | 59.4 | 60.5 | 60.2 | 58.0 | 60.3 | 60.6 | 60.3 | 60.6 | 60.4 | 60.3 |
| Jannis | 40000 | 54 | **78.6** | 78.0 | 75.0 | 77.2 | 73.8 | 73.9 | 77.4 | 78.4 | 74.6 | 78.6 | 75.1 | 74.1 |
| Bior | 2000 | 419 | 76.2 | 74.3 | 76.9 | 73.6 | 74.3 | 73.5 | 77.5 | 76.9 | 70.5 | **78.8** | 72.5 | 72.9 |
| Calif | 14000 | 8 | 88.8 | 88.1 | 87.2 | 86.6 | 82.1 | 84.7 | 86.6 | 85.4 | 84.5 | **89.0** | 85.3 | 85.5 |
| Heloc | 7000 | 22 | **71.9** | 71.6 | 66.2 | 71.3 | 70.2 | 70.7 | 71.4 | 71.3 | 68.6 | 71.1 | 71.2 | 68.4 |

We evaluate the DLGN(dlgn), DLGN-SF(dlgns), DLGN-VT(dlgnv), and DLGN-DT(dlgnd) algorithms, with width and depth hyperparameters chosen on a validation set. They are compared against the following standard algorithms.

**Non-tree algorithms**: *ReLU networks* (Fukushima, 1969): Classic multilayer ReLU activation neural network with depth and width as hyperparameters. *Disjunctive normal networks (disnn)* (Sajjadi et al., 2016): A neural network architecture explicitly designed to learn a union of the intersection of halfspaces, which exactly corresponds to the ODT labelling function with number of polytopes and number of half-spaces per polytope as hyperparameters. *Gated linear networks (gln)* (Veness et al., 2021): A backpropagation-free architecture where each unit (neuron) learns locally and models nonlinear functions using data-dependent gating (context) with layer sizes, input size and context map size as hyperparameters. *Support vector machines (SVM)* (Cortes & Vapnik, 1995): Constructs linear separators in a high dimensional space given by a non-linear kernel. The kernel choice and its parameters and the regularization constant are the key hyperparameters. *Linear SVM (svl)* corresponds to SVM with a linear kernel. *Soft decision tree SDT(sdt)* (Frosst & Hinton, 2017): A soft decision tree made out of a trained neural network. Although it is termed a decision tree, it gives a predictive distribution of classes at the leaf with the maximum path probability rather than explicitly dividing the data into left and right nodes recursively. It uses input dimension, output dimension, depth and the regularisation coefficient as hyperparameters.

**Tree algorithms**: *Classification and regression tree CART(cart)* (Breiman et al., 1984): It splits the data greedily in a parallel-axis fashion and uses maximum depth, minimum samples split and minimum samples leaf as main hyperparameters. *Standard random forest (rf)* (Ho, 1995): An ensemble method operating by constructing multiple axis-parallel decision trees at training. Number of trees estimators, maximum depth, minimum samples split and minimum samples leaf are its main hyperparameters. *Zan-DT(zan)* (Zantedeschi et al., 2020): A binary ODT learning algorithm that concur-

rently optimizes discrete and continuous parameters through sparse relaxation of a mixed-integer program, facilitating gradient flow for joint optimization with tree depth, regularization, number of layers and dropouts as hyperparameters. *Tree alternating optimization TAO(tao)* (Carreira-Perpinán & Tavallali, 2018): An ODT learning algorithm where the initial tree parameters are optimized by decreasing the misclassification error, using alternating optimization over node subsets that are separable with number of iterations, minimum sample nodes, minimum leaf samples and regularization parameters as hyperparameters.

Appendix A.10 details the experimental setup and all the hyperparameter sets used for these algorithms. The models exhibiting the highest performances are highlighted in **bold** in Tables 2 and 3, giving results on synthetic and real tabular datasets.

## 7.1 PERFORMANCE ON SYNTHETIC DATASETS

We construct three different synthetic datasets, where the data labels comes from COB-ODTs. The three datasets (SDI, SDII, and SDIII) have different input dimensions (20, 100, and 500, respectively) and use decision trees with depth 4. The dataset characteristics (number of training points $n$ and input dimension $d$) and results on these synthetic datasets are described in Table 2. Almost all the standard algorithms, including ODT learning algorithms, failed to perform satisfactorily on these datasets. Standard ReLU nets give good accuracy for some of these datasets, but they are sensitive to hyperparameter settings, particularly for higher dimensional data. Another qualitative property of DLGN-DT that is not fully reflected in Table 2 is that the hyperplanes in the learned tree closely match those in the true ODT. The main goal of designing DLGN and its variants is the study of feature learning, but it is also important to observe that they are indeed learning models with high accuracy and outperforming all other algorithms in some settings. The models exhibiting the highest and second highest performance are highlighted in **bold** and blue respectively in Tables 2.

## 7.2 PERFORMANCE ON TABULAR DATASETS

We assessed the performance of our methods, DLGN(dlgn) and DLGN-VT(dlgnv), compared to standard algorithms on real tabular datasets with the details of dataset characteristics (number of training points $n$ and input dimension $d$) outlined in Table 3. We used a total of 20 tabular binary classification datasets for the comparative study of our models. Most datasets are available in the UCI repository.[1] Some are taken from OpenML benchmark: (Grinsztajn et al., 2022).[2] We have performed the experiments with 5 different train test splits and the mean is considered here. The models exhibiting the highest performance are highlighted in **bold**, and models whose accuracy lie in 95 per cent confidence interval of the best performing model is marked by blue in Tables 3. We observe that the linear and kernel methods are almost always beaten by other algorithms which can potentially learn data-dependent features. DLGN and random forests are the best among the other algorithms. DLGN-VT also yielded competitive results while being more interpretable. This illustrates that feature learning is a useful property for these datasets, and DLGNs are capable of learning powerful features.

## 8 CONCLUSION

Feature learning in deep networks is a fundamental problem and this paper makes several advances on this problem – it gives a novel hypothesis, a novel data setup , a novel architecture and a way to verify if feature learning indeed happens during training. Exploiting this progress by getting better learning algorithms and theoretically proving that the proposed feature learning mechanism is present in training are exciting directions of future research. A further discussion of the feature learning narrative in this paper is given in the Appendix.

---

[1] https://archive.ics.uci.edu/datasets
[2] https://www.openml.org/search?type=benchmark&study_type=task&sort=tasks_included&id=298

ACKNOWLEDGEMENTS

HGR and CL thank the Robert Bosch Center for Data Science and Artificial Intelligence (RBCD-SAI) and the Wadhwani School of Artificial Intelligence (WSAI) at IIT Madras for its support.

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

## A APPENDIX

### A.1 DLGN ILLUSTRATIONS

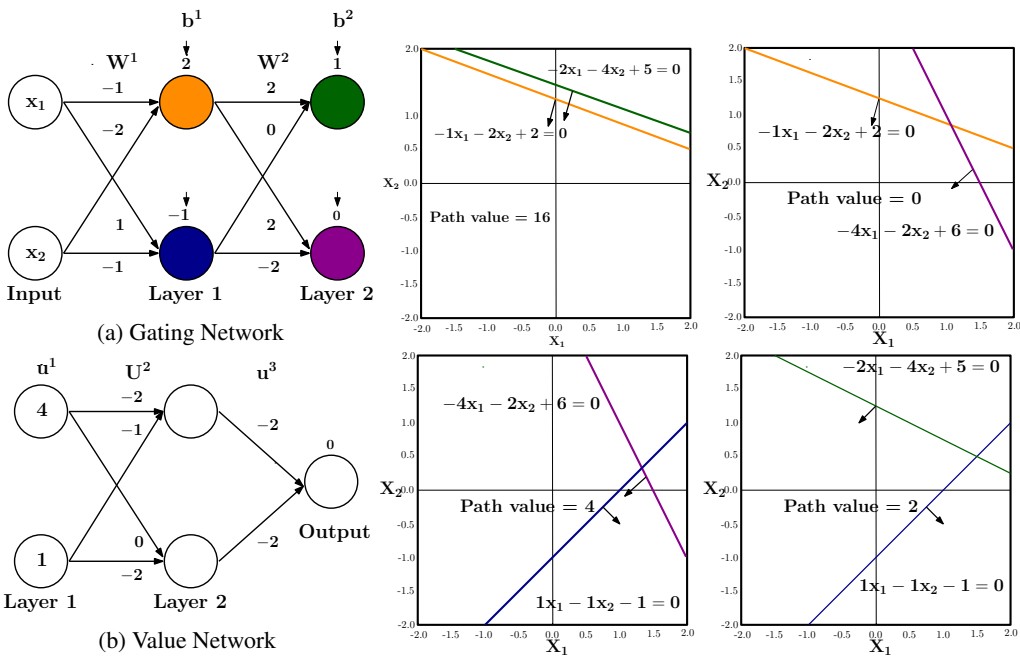

Figure 4: An illustration of DLGN network and node hyperplanes

Figure 4(a) and 4(b) give an example DLGN operating on a 2-dimensional input with 2 layers and 2 neurons per layer, i.e. $d = m = L = 2$. Figure 4(c) illustrates the region of activation for all the $m^L = 4$ paths given by the gating network 4(a). The value network in Figure 4(b) gives the path value for all these paths by multiplying the appropriate weights in the path.

### A.2 DLGN GATING HYPERPLANE CLUSTERING WITH LARGER WIDTH

Table 4: Number of DLGN hyperplanes (after training) within a given distance of the label function ODT hyperplanes. The 15 ODT internal nodes are numbered 0 to 14, with 0 as the root. The distance of the closest (initial and trained) DLGN hyperplane to all the ODT hyperplanes is given in the last 2 rows. The DLGN has 4 hidden layers with 100 neurons in each layer, i.e. $L = 4, m = 100$.

| Distance | 0 | 1 | 2 | 3 | 4 | 5 | 6 | 7 | 8 | 9 | 10 | 11 | 12 | 13 | 14 |
|---|---|---|---|---|---|---|---|---|---|---|---|---|---|---|---|
| 0.1 | 6 | 2 | 1 | 1 | 0 | 0 | 0 | 0 | 0 | 0 | 0 | 0 | 1 | 0 | 0 |
| 0.2 | 18 | 3 | 1 | 1 | 0 | 0 | 1 | 0 | 1 | 1 | 0 | 1 | 1 | 0 | 1 |
| 0.3 | 27 | 4 | 1 | 1 | 0 | 1 | 1 | 0 | 1 | 1 | 0 | 2 | 1 | 0 | 1 |
| Closest Dist.(init) | 1.16 | 1.2 | 1.19 | 1.21 | 1.11 | 1.21 | 1.14 | 1.19 | 1.20 | 1.15 | 1.18 | 1.20 | 1.14 | 1.20 | 1.18 |
| Closest Dist.(final) | 0.06 | 0.08 | 0.09 | 0.08 | 0.38 | 0.22 | 0.15 | 0.45 | 0.12 | 0.12 | 0.57 | 0.14 | 0.08 | 0.68 | 0.11 |

### A.3 DETAILS OF THE DLGN ARCHITECTURE VARIANTS

We also consider two natural variants of the DLGN architecture. The first is a simple re-parameterisation, where the gating network $f_\pi$ is parameterised directly by the matrices $V^1, V^2, \ldots, V^L$, each of shape $m \times d$, instead of using the parameters $W^1, \ldots, W^L$. Clearly, this reparameterisation does not lose any representation power. We call this parameterisation as DLGN-SF (for shallow features).

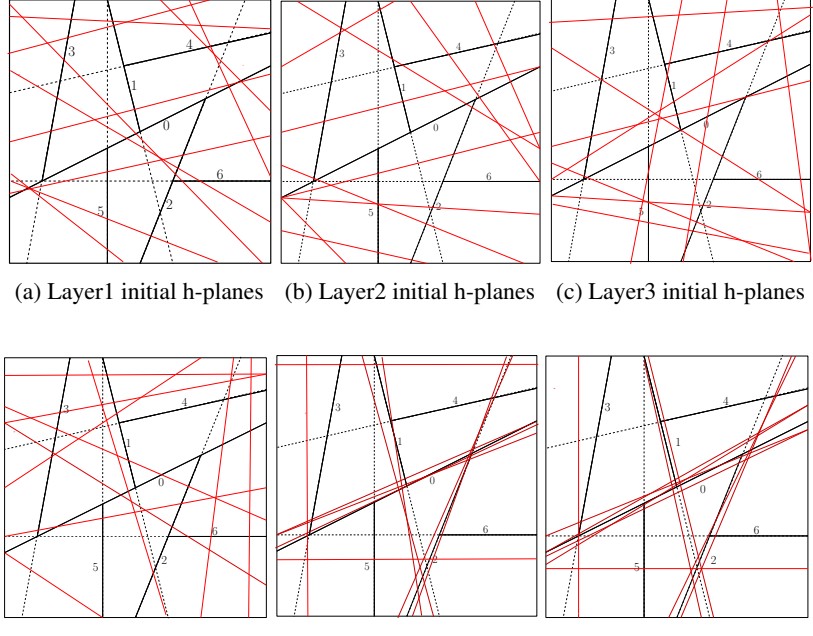

(a) Layer1 initial h-planes   (b) Layer2 initial h-planes   (c) Layer3 initial h-planes

(d) Layer1 trained h-planes   (e) Layer2 trained h-planes   (f) Layer3 trained h-planes

Figure 5: An illustration of DLGN hyperplanes before and after training on data in Fig 1c.

Table 5: Distance table for DLGN-SF Layers: 4 Nodes per layer: 10

| Distance | 0 | 1 | 2 | 3 | 4 | 5 | 6 | 7 | 8 | 9 | 10 | 11 | 12 | 13 | 14 |
|---|---|---|---|---|---|---|---|---|---|---|---|---|---|---|---|
| 0.1 | 1 | 1 | 1 | 1 | 0 | 0 | 0 | 1 | 0 | 0 | 1 | 0 | 0 | 0 | 0 |
| 0.2 | 1 | 1 | 1 | 1 | 1 | 1 | 1 | 1 | 1 | 0 | 1 | 1 | 1 | 1 | 0 |
| 0.3 | 1 | 1 | 1 | 1 | 1 | 1 | 1 | 1 | 1 | 1 | 1 | 1 | 1 | 1 | 1 |
| Closest Dist.(init) | 1.20 | 1.22 | 1.26 | 1.27 | 1.24 | 1.24 | 1.25 | 1.22 | 1.29 | 1.27 | 1.24 | 1.26 | 1.15 | 1.26 | 1.25 |
| Closest Dist.(final) | 0.05 | 0.07 | 0.08 | 0.08 | 0.12 | 0.11 | 0.13 | 0.09 | 0.18 | 0.26 | 0.08 | 0.16 | 0.10 | 0.14 | 0.21 |

Another variant that we use an explicit parameterisation of the coefficients $g_\pi$ instead of using a value network. This parameterisation is also clearly more powerful than the standard DLGN parameterisation in the previous section. However, this requires a parameter tensor of size $m^L$ and is not really practical for large $L$ and $m$. We call this reparameterisation as DLGN-VT (for value tensor).

All of these models can be extended to allow a bias parameter for the gating network so that the functions $\boldsymbol{\eta}^\ell$ can also be affine functions instead of strict linear functions, but we do not discuss this version here for the purpose of simplicity.

Number of DLGN hyperplanes (after training) within a given distance of the label function ODT hyperplanes of DLGN-VT and DLGN-SF are shown in Table 6 and Table 5. At initialization, all these numbers are equal to zero. The 15 ODT internal nodes are numbered 0 to 14, with 0 as the root. Results of similar experiments as DLGN (Table 1) shown in Table 5 and 6 for DLGN_SF and DLGN_VT respectively depicts identical hyperplane-seeking properties of these two variants of the DLGN architecture as well. For each node in the ODT, we count the number of DLGN hyperplanes within a distance of $0.1, 0.2$, and $0.3$ from it.

## A.4   LOCAL DISCONTINUITY COEFFICIENT ILLUSTRATION

Figure 6 gives an example of regions $\mathcal{R}_1, \mathcal{R}_2$ and $\mathcal{R}_3$ and manifold functions $f_1, f_2$. Based on the definition of the local discontinuity coefficient $\gamma$, we have the following. The manifold given by the root node $f_1(x) = 0$ has high $\gamma$ value in the smaller scope $\mathcal{R}_1$, but has a much smaller $\gamma$ value in the larger scope of $\mathcal{R}_2$. i.e. $\gamma(\mathcal{R}_1, f_1) \approx 1$ and $\gamma(\mathcal{R}_2, f_1) \approx 0$. The manifold given by any ODT internal

Table 6: Distance table for DLGN-VT Layers: 4 Nodes per layer: 10

| Distance | 0 | 1 | 2 | 3 | 4 | 5 | 6 | 7 | 8 | 9 | 10 | 11 | 12 | 13 | 14 |
|---|---|---|---|---|---|---|---|---|---|---|---|---|---|---|---|
| 0.1 | 3 | 2 | 2 | 0 | 2 | 2 | 1 | 1 | 0 | 1 | 2 | 1 | 1 | 1 | 1 |
| 0.2 | 3 | 2 | 2 | 0 | 2 | 2 | 1 | 1 | 0 | 2 | 2 | 1 | 1 | 1 | 1 |
| 0.3 | 3 | 2 | 2 | 0 | 2 | 2 | 1 | 1 | 1 | 2 | 2 | 1 | 1 | 1 | 1 |
| Closest Dist.(init) | 1.16 | 1.27 | 1.23 | 1.22 | 1.28 | 1.22 | 1.18 | 1.17 | 1.27 | 1.26 | 1.19 | 1.21 | 1.26 | 1.23 | 1.21 |
| Closest Dist.(final) | 0.03 | 0.05 | 0.06 | 0.35 | 0.08 | 0.09 | 0.06 | 0.09 | 0.24 | 0.06 | 0.04 | 0.07 | 0.05 | 0.07 | 0.05 |

$$f_1(\mathbf{x}) = \mathbf{u_1}^\top \mathbf{x} + b_1 = 0$$

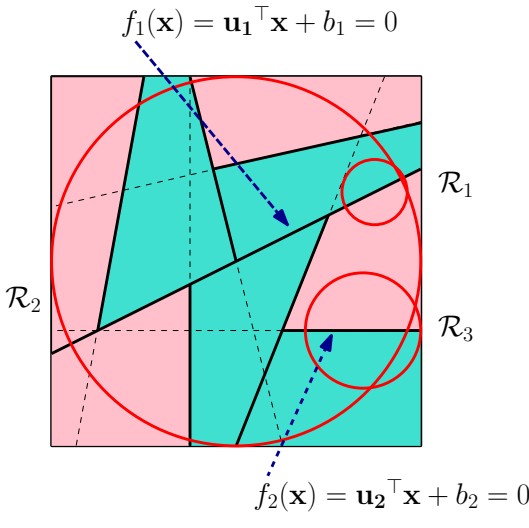

$$f_2(\mathbf{x}) = \mathbf{u_2}^\top \mathbf{x} + b_2 = 0$$

Figure 6: An illustration of Local Discontinuity Coefficient.

node hyperplane has a scope within which it is discontinuous, e.g. the scope of discontinuity of the manifold $f_2(\mathbf{x}) = 0$ would correspond to $\mathcal{R}_3$, i.e. $\gamma(\mathcal{R}_3, f_2) \approx 1$.

## A.5 PROOF OF DLGN COMPUTATION EQUIVALENCE

In this section we prove that Equation 1 is implemented by Equation 5.

For any $\ell \in [L]$, let $D^\ell(\mathbf{x})$ be an $m \times m$ diagonal matrix, whose $(i, i)^{\text{th}}$ entry is 1 if the $i^{\text{th}}$ hidden node in hidden layer $\ell$ is active and 0 otherwise, i.e.

$$[D^\ell(\mathbf{x})]_{i,i} = 1(\boldsymbol{\eta}_i^\ell(\mathbf{x}) \geq 0)$$

where $\boldsymbol{\eta}$ is the gating function defined in Equation 3. We suppress the dependence of $D^\ell$ on $\mathbf{x}$ for convenience below. Consider the following function of $\mathbf{x}$ :

$$\widetilde{y}(\mathbf{x}) = (\mathbf{u}^{L+1})^\top D^L U^L D^{L-1} \ldots D^2 U^2 D^1 \mathbf{u}^1 = (\mathbf{u}^{L+1})^\top \left( \prod_{\ell=L}^{2} D^\ell U^\ell \right) D^1 \mathbf{u}^1 \qquad (6)$$

We will show that the above expression is equal to the RHS of equation 1 when the path values $g_\pi$ are represented as a product of pairwise terms as in Equation 4.

Writing the equation above as product of $L+1$ matrices, with the first and last being vectors of size $m$ and the intermediate matrices $(D^\ell U^\ell)$ as matrices of size $m \times m$, and expanding the terms of the matrix product as a sum of $m^L$ terms

$$\widetilde{y}(\mathbf{x}) = \sum_{(i_L, i_{L-1}, \ldots, i_1) \in [m]^L} \left( \mathbf{u}_{i_L}^{L+1} \prod_{\ell=L}^{2} \left( D_{i_\ell, i_\ell}^\ell U_{i_\ell, i_{\ell-1}}^\ell \right) D_{i_1, i_1}^1 \mathbf{u}_{i_1}^1 \right)$$

$$= \sum_{(i_L, i_{L-1}, \ldots, i_1) \in [m]^L} \left( \prod_{\ell=1}^{L} D_{i_\ell, i_\ell}^\ell(\mathbf{x}) \right) \left( \mathbf{u}_{i_1}^1 \prod_{\ell=2}^{L} \left( U_{i_\ell, i_{\ell-1}}^\ell \right) \mathbf{u}_{i_L}^{L+1} \right) = \sum_{\pi \in [m]^L} f_\pi(\mathbf{x}) g_\pi$$

To see that the RHS of Equation 6 also corresponds to the RHS of equation 5, just observe that $h^k(\mathbf{x}) = \prod_{\ell=k}^2 \left( D^\ell(\mathbf{x}) U^\ell \right) D^1(\mathbf{x}) \mathbf{u}^1$. Thus the computational procedure given by Equation 5, exactly implements the conceptual path decomposition given in Equation 1.

### A.6 RELATION BETWEEN THE ReLU NETWORK AND THE DLGN

While the DLGN might seem like a completely different architecture from the ReLU network, it is quite intimately related to and motivated from the ReLU network.

The ReLU network can also be expressed in a matrix product format like Equation 6. However, the elements of the diagonal matrices $D^\ell$ (except the first layer with $\ell = 1$) are not simple halfspace indicator functions. They still correspond to whether the corresponding node is 'active' or 'not active'.

In this case, we can still view elements of the matrices $D^\ell$ as features of the network, which corresponds to the standard usage of features as the activation maps of neurons. The ReLU network features would correspond to 'bent hyperplanes'. However, the simple view of connected but separate features that the DLGN affords is no longer viable. We currently have no insight into why bent hyperplane features could be better than straight hyperplanes.

### A.7 VALUE OF THE FEATURE LEARNING NARRATIVE FROM DLGN

We believe that the feature learning insights drawn from DLGN are applicable in general. More importantly, we believe that it can shed light on several phenomena that remain mysteries with deep networks. Some of the mysteries and candidate answers motivated from this paper are given below.

Q: Is neural network training using gradient descent parameter efficient?

A: No. The greedy nature of gradient descent, pushes multiple interconnected parts towards the same feature when it is potentially not required. e.g. In Table 4 the distances of learned DLGN hyperplanes (with width $m = 100$) to the ODT hyperplanes are given. While the root node hyperplane is discovered separately by multiple gating hyperplanes, some of the other internal nodes are left in the lurch. This causes poor generalisation. Potentially, this narrative can be exploited to make gradient descent less greedy.

Q: How does gradient descent discover the label function discontinuities in high dimensional input space?

A: The full labeling function may be far from being a linear separator, but when data is restricted within a small enough scope, one of the manifolds making up the decision boundary could very well be a good classifier (this argument uses the insight that even in high-dimensional data, if it is linearly separable, picking up the separator can be done in a compute and data efficient manner). This causes that manifold to be picked up by one of the components of the deep network, and thereby making the rest of the learning problem easier and kickstarting a virtuous cycle.

Q: Why does neural network pruning enable better learning of smaller architectures than training the smaller architecture from scratch?

A: We assume effective learning happens only in problems where the number of label function discontinuities is small. Significant parts of the neural network are simply not necessary to represent the discontinuities, which can be done by a small fraction of the trained deep network model. A large width network is still better for training, because (say) doubling the number of neurons per layer increases the number of paths by a factor of $2^L$. This increases the chances of some path picking a right scope during training enabling the learning of the appropriate label function discontinuity. Once the learning is complete, a significant fraction of the network which were unlucky to not have gotten a good scope can be removed.

Q: What is the role of layers in deep networks? Is increasing the number of layers always beneficial?

A: The main role of layers is to give context/scope to other layers. For example, with a depth 4 ODT labelling function, a DLGN with 3 or lesser number of hidden layers would not be able to give a good scope to any neuron. A DLGN with 5 or more layers is just unnecessary.

Table 7: Synthetic Datasets Generation

| dataset | total_samples | train_samples(n) | n_features(d) | depth | seed | thres | generate_data |
|---------|---------------|------------------|---------------|-------|------|-------|---------------|
| SDI | 40000 | 20000 | 20 | 4 | 365 | 0 | through code |
| SDII | 60000 | 30000 | 100 | 4 | 365 | 0 | through code |
| SDIII | 100000 | 50000 | 500 | 4 | 365 | 0 | through code |

This is reflected in our experiments as well, where depth 4 DLGN performed best for a depth 4 ODT labelling function.

## A.8 DETAILS OF SUBROUTINES USED IN DLGN-DT

This function takes into account the DLGN model after training. It iteratively calculates the effective weights of the layer using the formula $V^\ell = W^\ell W^{\ell-1} \ldots W^1$ as in lines 3 of the algorithm. It finally returns each layer's effective weight (hyperplanes) as a vector.

---

**Algorithm 3** Return gates of a trained DLGN model

---

**Arguments:** A DLGN model with parameters $W^1, \ldots, W^L, \mathbf{u}^1, U^2, \ldots, U^L, \mathbf{u}^{L+1}$
**Outputs:** $mL$ hyperplanes in the input dimension

1: **function** GATEHYPERPLANES(`model`)
2:      **for** $l \leftarrow 1$ **to** $L$ **do**
3:          $V^\ell \leftarrow W^\ell W^{\ell-1} \ldots W^1$
4:      **end for**
5:      **return** $V \leftarrow V^1, \ldots, V^L$
6: **end function**

---

## A.9 DATASETS USED

We have used 3 synthetic datasets and 20 tabular datasets to evaluate the performance of our model DLGN and its variants against some standard algorithms. In this section, we will elaborately describe the datasets used.

### A.9.1 SYNTHETIC DATASETS:

This dataset is synthetically generated with specified dimensions from a labelling function $f^*$ given by an Oblique Decision Tree (ODT) with depth and a defined number of data points as given in Table 7. The datapoints $\mathbf{x}$ are drawn uniformly from the surface of a $d$-dimensional sphere of radius 1, centred at the origin. We used COB-ODTs as mentioned in Section 3 with biases kept at zero. The leaf node labels are chosen so that sibling labels get opposite signs. The final output includes the pruned data, labels, and information about the tree's structure. Three synthetic datasets (SD) are used, named SDI, SDII, and SDIII.
Table 7 presents the parameters used for constructing the datasets.

### A.9.2 TABULAR DATASETS:

We used a total of 20 tabular binary classification datasets for the comparative study of our models. Most datasets are available in the UCI repository `https://archive.ics.uci.edu/datasets`. Some are taken from the OpenML benchmark, as given in the paper (Grinsztajn et al., 2022). `https://www.openml.org/search?type=benchmark&study_type=task&sort=tasks_included&id=298`. The dataset download URL is in Table 8. After downloading the datasets, they are preprocessed by dropping rows with missing values, converting categorical features using Label Encoding, Standardizing numerical features, and Encoding the target variables to 0 and 1.

Figure 5 illustrates an example scenario when a 3-hidden layer DLGN is trained on data given in Figure 1(c). The initial hyperplanes(h-planes) given by $V^1$ and $V^2$ and $V^3$ as shown in Figure 5(a-c)

Table 8: Tabular datasets

| data | download_link |
|------|---------------|
| Adult | `https://archive.ics.uci.edu/static/public/2/adult.zip` |
| Bank | `https://archive.ics.uci.edu/static/public/222/bank+marketing.zip` |
| Card | `https://archive.ics.uci.edu/static/public/350/default+of+credit+card+clients.zip` |
| Telesc | `https://archive.ics.uci.edu/static/public/159/magic+gamma+telescope.zip` |
| Rice | `https://archive.ics.uci.edu/static/public/545/rice+cammeo+and+osmancik.zip` |
| Stat | `http://archive.ics.uci.edu/static/public/144/statlog+german+credit+data.zip` |
| Spam | `http://archive.ics.uci.edu/static/public/94/spambase.zip` |
| Gyro | `https://archive.ics.uci.edu/static/public/755/accelerometer+gyro+mobile+phone+dataset.zip` |
| Swar | `https://archive.ics.uci.edu/static/public/524/swarm+behaviour.zip` |
| Credit | `https://api.openml.org/data/v1/download/22103185/credit.arff` |
| Elec | `https://api.openml.org/data/v1/download/22103245/electricity.arff` |
| Cover | `https://api.openml.org/data/v1/download/22103246/covertype.arff` |
| Pol | `https://api.openml.org/data/v1/download/22103247/pol.arff` |
| House | `https://api.openml.org/data/v1/download/22103248/house_16H.arff` |
| Mini | `https://api.openml.org/data/v1/download/22103253/MiniBooNE.arff` |
| Diab | `https://api.openml.org/data/v1/download/22111908/Diabetes130US.arff` |
| Jannis | `https://api.openml.org/data/v1/download/22111907/jannis.arff` |
| Bior | `https://api.openml.org/data/v1/download/22111905/Bioresponse.arff` |
| Calif | `https://api.openml.org/data/v1/download/22111914/california.arff` |
| Heloc | `https://api.openml.org/data/v1/download/22111912/heloc.arff` |

are essentially random. However, after training, the hyperplanes in the later layers show a remarkable tendency to move towards the hyperplanes corresponding to the decision tree – particularly that of nodes close to the root (See Figures 5(d-f)).

## A.10 EXPERIMENTAL SETUP DETAILS

### A.10.1 TRAIN_VALIDATION_TEST SPLIT:

For the synthetic datasets SDI, SDII, and SDIII, the dataset is split into 50% train, 25% test, and 25% validation set. Models are trained on the training data and validated on the validation set, and then the test score is reported against the test data with the best hyperparameters. Similarly, for the tabular datasets, the dataset is split into 60% train, 20% test, and 20% validation set.

A.10.2   NUMBER OF FOLDS:

Based on the algorithms used, the number of folds used is also varied. For standard ML algorithms like CART, Random Forest, SVM, and SVM Linear, we use 3-fold cross-validation; for most other algorithms, including DLGNs, we use one-fold size.

A.10.3   HARDWARE:

All the experiments are performed on Kaggle, and all the neural network-based experiments use GPU, whereas traditional ML algorithms are performed on CPU. Kaggle provided GPUs, such as GPU T4 x2 and GPU P100.

A.10.4   HYPERPARAMETERS TUNING:

Each algorithm used in this paper has a different set of hyperparameters, and hyperparameter tuning is one of the most important aspects for getting the best accuracy. Here, for each algorithm, we extensively searched from a collection of hyperparameters and validated their result on the validation set to obtain the best-performing hyperparameters. Test results are reported based on the best hyperparameters. The below tables give a list of all hyperparameters for each algorithm used.

Table 9: DLGN and DLGN-SF hyperparameters space

| Parameters | Set of values searched on |
|---|---|
| num layers | 3, 4, 5 |
| num nodes(each layer) | 10, 20, 50, 100, 200, 500, 1000 |
| beta | 3, 10, 20, 30 |
| learning rate(lr) | 0.1, 0.01, 0.02, 0.05, 0.001, 0.005, 0.0001 |
| epochs | 200, 300, 500 |
| batches | 1, 10, 100 |
| optimizers | Adam, SGD |
| weight decay | 0, 0.1 |

Table 9 lists the hyperparameters for training DLGN and DLGN-SF across all datasets. The best combination, selected via validation, determines test accuracy. Key parameters include num layers, num nodes (per layer), and beta.

Table 10: DLGN-VT hyperparameters space

| Parameters | Set of values searched on |
|---|---|
| num layers | 3, 4 |
| num nodes(each layer) | 10, 20 |
| beta | 3, 10, 20, 30 |
| learning rate(lr) | 1., 0.1, 0.01, 0.02, 0.05, 0.001, 0.005, 0.0001 |
| epochs | 200, 300, 500,1000,5000,10000,15000 |
| batches | 1, 10, 100 |
| optimizers | Adam, SGD |
| weight decay | 0, 0.1 |
| C | 0.1, 0.03, 1.0 |
| max_iter | 100, 500, 1000 |
| penalty | l1, l2 |
| solver | liblinear |

Table 10 lists the hyperparameters for training DLGN-VT across all datasets. The best combination, selected via validation, determines test accuracy. Along with the DLGN parameters, it has C, max_iter, penalty and solver as additional parameters.

Table 11: DLGN-DT hyperparameters space

| Parameters | Set of values searched on |
|---|---|
| num layers | 3, 4, 5 |
| num nodes(each layer) | 10, 20, 50, 100, 200, 500, 1000 |
| beta | 3, 10, 20, 30 |
| learning rate(lr) | 0.1, 0.01, 0.02, 0.05, 0.001, 0.005, 0.0001 |
| epochs | 200, 300, 500 |
| batches | 1, 10, 100 |
| optimizers | Adam, SGD |
| weight decay | 0, 0.1 |
| eps | 0.1, 0.2, 0.3 |
| min_samples | 1, 2, 3, 5, 7, 10, 15, 22, 40 |
| max_depth | 1 to 10 |

Table 11 lists the hyperparameters for training DLGN-DT across all datasets. The best combination, selected via validation, determines test accuracy. Key parameters include eps, min_samples, and max_depth for clustering and tree depth.

Table 12: ReLU hyperparameters space

| Parameters | Set of values searched on |
|---|---|
| num layers | 3, 4, 5 |
| num nodes(each layer) | 10, 20, 50, 100, 200, 500, 1000 |
| learning rate(lr) | 0.1, 0.01, 0.02, 0.05, 0.001, 0.005, 0.0001 |
| epochs | 200, 500, 1000 |
| optimizers | Adam, SGD |
| weight decay | 0, 0.1 |

Table 12 lists the hyperparameters for training ReLU models across all datasets. The best combination, selected via validation, determines test accuracy.

Table 13: SVM Linear hyperparameters space

| Parameters | Set of values searched on |
|---|---|
| C | 0.1, 0.5, 1, 2, 5 |
| kernel | Linear |

Table 13 lists the hyperparameters for training Linear SVM across all datasets. The best combination, selected via validation, determines test accuracy, with the kernel set to Linear.

Table 14 lists the hyperparameters for training Non-linear SVM across all datasets. The best combination, selected via validation, determines test accuracy, with the kernel set to rbf or sigmoid.

Table 15 lists the hyperparameters for training the CART model across all datasets. The best combination, selected via validation, determines test accuracy, with max_depth as the key parameter.

Table 16 contains the hyperparameter set used in training the random forest model on all the datasets. The trained model is tested on a validation set, and the best hyperparameter combination is found, which is then used to get the test accuracy. n_estimators define the number of estimators in the random forest as one of the most vital hyperparameters.

Table 17 contains the hyperparameter set used in training the SDT model on all the datasets. The trained model is tested on a validation set, and the best hyperparameter combination is found, which is then used to get the test accuracy. depth is the tree depth of SDT which is the most vital hyperparameter to tune.

Table 14: SVM hyperparameters space

| Parameters | Set of values searched on |
|---|---|
| C | 0.1, 0.5, 1, 2, 5 |
| kernel | rbf, sigmoid |
| gamma | scale, auto, 0.001, 0.01, 0.1, 1, 10 |
| degree | 2, 3, 4, 5 |

Table 15: CART hyperparameters space

| Parameters | Set of values searched on |
|---|---|
| criterion | gini, entropy |
| splitter | best, random |
| max_depth | 1 to 10 |
| min_samples_split | 1 to 10 |
| min_samples_leaf | 1, 2, 4, 5 |
| max_features | sqrt, log2 |

Table 16: Random Forest hyperparameters space

| Parameters | Set of values searched on |
|---|---|
| criterion | gini, entropy |
| n_estimators | 10, 20, 50, 100 |
| max_depth | 1 to 10 |
| min_samples_split | 1 to 10 |
| min_samples_leaf | 1, 2, 4, 5 |
| max_features | sqrt, log2 |

Table 17: SDT hyperparameters space

| Parameters | Set of values searched on |
|---|---|
| depth | 1 to 10 |
| lamda | 0.1, 0.01, 0.02, 0.05, 0.001 |
| learning rate(lr) | 0.1, 0.01, 0.02, 0.05, 0.001, 0.005, 0.0001 |
| epochs | 200, 300, 500 |
| batches | 32, 64, 128 |
| weight decay | 0, 0.1, 0.0005 |

Table 18: TAO hyperparameters space

| Parameters | Set of values searched on |
|---|---|
| n_iters | 10, 20, 30 |
| max_leaf_nodes | 5, 10, 15 |
| randomize_tree | True, False |
| update_scoring | accuracy |
| min_node_samples_tao | 1, 2, 3, 4, 5 |
| min_leaf_samples_tao | 1, 2, 3, 4, 5 |
| reg_param | 0.1, 0.01, 0.02, 0.05, 0.001 |

Table 18 contains the hyperparameter set used in training the TAO model on all the datasets. The trained model is tested on a validation set, and the best hyperparameter combination is found, which is then used to get the test accuracy. n_iters, max_leaf_nodes and min_node_samples are important hyperparameters to train.

Table 19: Zan-DT hyperparameters space

| Parameters | Set of values searched on |
| --- | --- |
| depth | 1 to 10 |
| reg | 0.1, 0.01, 1.48, 1.5, 2 |
| mlp_layer | 3, 4, 5 |
| dropout | 0.0, 0.01, 0.05, 0.07, 0.1 |
| lr | 0.1, 0.01, 0.02, 0.05, 0.001 |
| epochs | 200, 300, 500 |
| batches | 32, 64, 128, 512 |

Table 19 contains the hyperparameter set used in training the Zan-DT models on all the datasets. The trained model is tested on a validation set, and the best hyperparameter combination is found, which is then used to get the test accuracy. depth, reg and mlp_layer are vital parameters.

Table 20: Disnn hyperparameters space

| Parameters | Set of values searched on |
| --- | --- |
| n_polytopes_list | 1 - 15,20,30,100 |
| m_list | 1 - 15,20,30,100 |

Table 20 contains the hyperparameter set used in training the Disnn model on all the datasets. The trained model is tested on a validation set, and the best hyperparameter combination is found, which is then used to get the test accuracy. A value of 10-15 for n_polytopes_list and m_list works best.

Table 21: GLN hyperparameters space

| Parameters | Set of values searched on |
| --- | --- |
| layer_sizes | 3,4,5 |
| num_nodes | 5,10,20,50 |
| context_map_size | 2 to 10 |
| lr | 0.1, 0.01, 0.02, 0.05, 0.001,0.00025 |

Table 21 contains the hyperparameter set used in training the GLN model on all the datasets. The trained model is tested on a validation set, and the best hyperparameter combination is found, which is then used to get the test accuracy. layer_sizes, num_nodes and context_map_size are important parameters. context_map_size = 4 gives the best result.

