# OpenReview forum: "Deep Networks Learn Features From Local Discontinuities in the Label Function"
_ICLR.cc/2025/Conference — ICLR 2025 Poster_

### Official Review · Reviewer_xQ4v · 2024-10-29

**Soundness:** 2
**Presentation:** 3
**Contribution:** 3
**Rating:** 8
**Confidence:** 2

**Summary:**

This paper proposes a theory to explain how feature learning happens in neural networks. The authors posit that neural networks align (or rather, get attracted to) the discontinuites in the way the target label changes over the input domain. As a pilot test for this theory, the authors consider a setting where the target function to learn is an Oblique Decision Tree (ODT). ODTs correspond to decision trees where each internal node is a linear threshold. Thus, the hyperplanes associated to the nodes in the decision tree split up the input space into different labeled regions (see Figure 1). The hyperplane corresponding to the root is "most discontinuous" in terms of how the label changes on both sides of it (owing to further and further splits by its many descendants). The authors posit that the training of non-linear neural networks procedurally aligns the model with these discontinuities.

As a tractable model to further empirically study this theory, the authors propose a novel neural network architecture which they term Deep Linearly Gated Networks (DLGNs). DLGNs are somewhere in between standard nonlinear (e.g., ReLU activated) deep networks and deep linear networks. The function computed by a DLGN can be writen as a large summation of terms of the form $f_\pi \cdot g_\pi$. Here, $f_\pi$ is a product of the signs of activations of neurons in a deep linear network (a single neuron from each layer). $g_\pi$ is a product of the activations themselves (a slight detail is that the weight matrices producing the activations in $f_\pi$ and $g_\pi$ are different). Thus, we can think of $f_\pi$ as an indicator of the intersection of halfspaces, while $g_\pi$ is the weight we add up if the indicator turns out to be true.

The authors fix the target function to be an ODT, and train the DLGN architecture on synthetic data labelled by the ODT. The suprising observation is given in Figure 2. At the end of training, if one plots the hyperplanes corresponding to the linear threshold at every neuron, one observes that the hyperplanes (at least in the later layers of the DLGN) align very well with the hyperplane splits in the ODT! Furthermore, Table 1 shows that most of these linear thresholds align with some hyperplane in the ODT. That is, the neurons in the DLGN architecture are "getting attracted" to aligning with some internal node in the ODT.

The authors next use this empirical observation to extract a decision tree out of a trained DLGN. Namely, they plot the linear thresholds corresopnding to all the activations after training, and then cluster these thresholds. The center of the largest cluster is chosen to be a root node hyperplane, and then we recurse this procedure on data on either side of this hyperplane. This procedure is pictorally well illustrated in Figure 3. Again, one can see that on performing such clustering-based decision tree generation, the first cluster center aligns very well with the root node hyperplane, and the phenomenon continues as we recurse. The upshot is that one can extract an interpretable model from the DLGN..

Finally, the authors perform experiments on real-world classification data to illustrate that the classification accuracy of DLGNs is somewhere in between standard ReLU networks and other simpler non-neural network algorithms. Thus, DLGNs have the added benefit of being interpretable, but more powerful than stanard decision trees.

**Strengths:**

The proposed theory is intriguing and indeed very interesting. The plots in Figures 2 and 3 are indeed striking---they illustrate how faithfully the DLGN neuron activations are increasingly aligning with the ODT hyperplanes over the process of training. These observations support the theory put forth by the authors about neural network architectures possibly picking up on the discontinuities in the labelling function. The clustering procedure to extract a decision tree is also very interesting, and empirically seems to work well (as suggested by Figure 3). Overall, I find the theory proposed by the authors, along with the striking empirical illustrations, very interesting. These could motivate further theoretical investigations for such phenomena.

**Weaknesses:**

The pilot experiments done by the authors are admittedly specialized. Namely, they don't actually consider standard ReLU networks, but only the DLGNs they propose. Furthermore, they only consider cases where the target function is conveniently an ODT. While this is totally okay as an initial starting point, it does raise the question about whether such empirical phenomena of the neurons aligning with the discontinuities also arise in cases where the target function has discontinuities of a different nature (like curvy discontinuities, etc). But this is not a significant weakness, as it seems beyond the scope of a pilot study. But it would be really interesting to visualize the activations of the neurons (say with quadratic thresholds) for when the target function is also composed of curvy discontinuities.

**Questions:**

1) Why do you even introduce a manifold $M$ in your notation in line 133? It is never used
2) On line 192, I don't think I agree that "no other hyperplane other than the internal nodes has this property". e.g. consider any other hyperplane that is not one of the internal nodes in the plot in Figure 1; doesn't it also have points of both labels on either side of it?
3) You introduce this formal notion of $\gamma(R, f)$ as the local discontinuity coefficient in Section 3, but then never mention it anywhere later in the paper. How would you explain the results in say Figure 2 and 3 in the context of this quantity? Could you elaborate on this a bit?

---

> ### Author Response · Authors · 2024-11-16
> **Response to reviewer xQ4v**
>
> We thank the reviewer for the helpful feedback. Response to questions below. Points 1 and 2 are for question 1, points 3,4 and 5 are for questions 2 and 3.
>
> 1. In a fuller version of the paper, we did have a paragraph connecting the manifold to DL architectures which we removed for brevity. The manifold still exists in the current version for reasons of generality. The definition of label discontinuties is applicable to any manifold while we study only hyperplanes. The DLGN model discontinuities are indeed just hyperplanes, but the geometry and parameterisation of the discontinuity manifold in a model is one of the core architecture design principles -- e.g. ReLU networks have "bent hyperplane" discontinuity manifolds, and CNNs have a more complex bouquet of bent hyperplanes instead of a single bent hyperplane due to the parameter tying and convolutional nature of the weights. We do not have any immediately insightful comments on why these structures should give better models on real data than simple hyperplanes like DLGNs.
>
> 2. Conceptually, we could have DQGN (deep quadratic gated network) in which the gating function for each neuron is an indicator of a quadratic function instead of a linear function. We could do that if we have a strong belief that the true label function can be compactly represented by just a few piecewise quadratic discontinuties instead of (say) millions of piecewise linear boundaries.
>
> 3. Fair point regarding other hyperplanes. We need to restrict the region $\mathcal{R}$ to reasonable sets for this to make sense. e.g. we should only consider those sets $\mathcal{R}$ that are contiguous and intersect with both sides of the manifold $f(x) =0 $. Otherwise the conditional expectation in the expression would  be vacuous. Under such reasonable restrictions, only the internal node hyperplanes (given by some f(x)=0) have non-trivial regions $\mathcal{R}$ such that $\gamma(\mathcal{R}, f) = 1$. Informally, it says that any input $x \in \mathcal{R}$ and a slightly perturbed input $x+\delta$ have different labels. The perturbation $\delta$ is chosen normal to the manifold $f$. We will fix this issue in the revised version.
>
> 4. The definition of $\gamma(\mathcal R, f)$ was given for the purpose of quantifying the level of different discontinuities in the label function. e.g. all the internal node hyperplanes in an ODT labelling function are clearly label function discontinuties in the informal sense, but how does one quantify the intuition that the root node hyperplane is a bigger discontinuity than a leaf node hyperplane? So we bring in a notion of "scope" of discontinuity. We define the scope of a discontinuity manifold $f$ to be the largest region $\mathcal{R}$ such that $\gamma(\mathcal{R}, f) = 1$. Informally, it is the region in the input space in which an input $x$ and a slightly perturbed input $x+\delta$ with perturbation $\delta$ normal to the manifold $f$ have different labels. Clearly, the root node hyperplane has a larger discontinuity scope than the leaf node hyper planes. We will add a figure illustrating this in the appendix. Also, this intuition really makes sense only in higher dimensions and with COB-ODTs, where most pairs of hyperplanes are normal to each other and hence the figure (which is in 2d and does not correspond to an ODT with orthogonal node hyperplanes) could be misleading.
>
> 5. The reason we even study discontinuity scopes is because of an empirical observation. In large trained DLGN architectures with COB-ODT labelling functions, disproportionately many DLGN gating hyperplanes go to the ODT root node hyperplane, while only a few of them move towards the leaf node hyperplanes.

---

> > ### Comment · Reviewer_xQ4v · 2024-11-21
> > **Response**
> >
> > Thank you for your response. I maintain that I find the empirical observations interesting and of value, and hence I will maintain my score.

---

### Official Review · Reviewer_n82d · 2024-11-02

**Soundness:** 2
**Presentation:** 2
**Contribution:** 3
**Rating:** 6
**Confidence:** 4

**Summary:**

In this paper, the authors propose a mechanism to intuitively explain why neural networks can surpass kernel methods through feature learning. They hypothesize that the feature learning process involves the alignment of model function discontinuities with label function discontinuities during training. To explore this, they introduce a new network architecture called the Deep Linearly Gated Network (DLGN), designed as a surrogate for ReLU networks. They argue that this architecture retains similarities to ReLU networks while offering easier interpretability. Under this framework, they provide empirical evidence showing how model function discontinuities move toward label function discontinuities during training, facilitated by feature learning.

**Strengths:**

This paper is a bold attempt to deepen our understanding of feature learning in deep learning. The hypothesis, data setup, network architecture, and approach to interpretability are unconventional, and they bring a fresh perspective to the literature. This work has the potential to inspire new ideas and serve as a valuable starting point for further exploration.

**Weaknesses:**

From my perspective, while this work is intriguing, it is not yet ready for formal publication. My concerns are as follows:

1. The authors reference previous studies examining the dynamics of single hidden layer models under specialized data and settings to push beyond kernel methods or deep linear models (Damian et al., 2022; Ba et al., 2022). They appropriately note that these analyses, often focused on specific data settings like the parity function, fall short of addressing the needs or behaviors of deeper networks. However, numerous works also investigate complex feature learning with deep neural networks, such as https://arxiv.org/abs/2305.06986 and https://arxiv.org/pdf/2311.13774. These studies should be acknowledged and compared with the current work.

2. I am skeptical about the extent to which the new architecture resembles a ReLU network. Additionally, it is unclear how this intuition or design could extend to CNNs or transformers. Besides, the current version only applies to binary classification, whereas a universal feature learning mechanism should ideally apply across setups, such as binary classification, multi-class classification, and regression. I am unsure how the proposed intuition extends to these broader contexts. For example, existing papers (such as the two mentioned above) demonstrate that neural networks can efficiently learn $h = g \circ p$ with $p$ quadratic and $g$ nonlinear via feature learning. How would this intuition explain such cases?

3. In Section 3, the introduction to ODT progresses too quickly. The paper would benefit from a more mathematically detailed introduction to the new concepts presented in this section.

**Questions:**

See the weakness part.

---

> ### Author Response · Authors · 2024-11-17
> **Response to reviewer n82d (Part 1 of 2)**
>
> We thank the reviewer for the feedback.
>
> 1. Thanks for the references, we will add a line discussing these as well in the revised version. These would squarely fall in the family of recent feature learning literature that aims to go beyond NTK.
>
> 2. **Link between DLGN and DNN** : The tale behind the birth of the DLGN is rooted in the ReLU network. A ReLU network can be represented as $  y(x) = W_L* D_L* \ldots W_1* D_1 * W_0 * x $. Where the matrices $D_i$ are diagonal matrices that depend on $x$, taking values 1 or 0 depending on whether the corresponding neuron is active for the input $x$. When viewed as a function of $x$ the $i^{th}$ diagonal element of the matrix $D_l$ is rather complex, except for $l=1$ where it is simply equal to 1 on a halfspace and 0 outside it. DLGN simply makes the gating function for every neuron as a halfspace, whose parameters are given separately, i.e. in the above ReLU model output expression replace the diagonal matrices $D_\ell$ with diagonal matrices containing $\eta^\ell$ as defined in the paper.
>
> 3. **Other architectures**: Studying CNNs or transformers is beyond the scope of the current work, but in other work we have been able to adapt multiclass CNNs to DLGNs and get performance within a few percentage points of a comparable ReLU net on CIFAR datasets. We reiterate however that the goal of this paper is to initiate discussion and inspire experiments/hypotheses of learning mechanisms that would not even be possible with ReLU networks. (see response to reviewer tLyw for more details). Pasting a paragraph from that response here for convenience.
>
> 4. **New lines of attack on feature learning** : The halfspaces in the gating network seeking out the label function discontinuities (as demonstrated in Table 1) gives us a way to ask questions regarding parameter efficiency etc. Clearly, there is no need for multiple DLGN halfspaces to go towards the same discontinuity, but gradient descent (which is a greedy procedure) does it anyway. Maybe it would be possible to exploit this and have learning routines go beyond the greedy nature of gradient descent and build smaller models that perform similarly (say, identify two different gates going towards the same halfspace and pushing away one of them). This is however beyond our scope right now, but the very fact that such discussions are possible is the main point of this paper.
>
> 5. **Other contexts** : The main claim of model features seeking out label function discontinuities is general enough to handle other contexts in supervised learning. While, the notion of label function discontinuities would still be sensible for multi-class classification, we will have to generalize this to include "high-slope" regions for regression. For example, consider the paper "Learning Hierarchical Polynomials with Three-Layer Neural Networks" by Wang et al. where the label function has the form $h=g \circ p$. Let the scalar function $g$ be such that its graph is mostly flat but has jumps at values 1,2,3. Then the label function discontinuties exactly correspond to the manifold p(x)=1, p(x)=2 and p(x)=3. The message in our paper would suggest that these manifolds will also appear in a trained deep network, which is indeed supported by a concrete theorem in Wang et al.
>
> 6. **Message surprise** : In a sense as mentioned by reviewer tLyw this observation is not surprising -- any model that generalises has to discover these discontinuities in some form or the other. The complexity of the ReLU network architecture makes it so that one is unable to figure out where the representation of these discontinuities is hidden in a trained high accuracy model. The DLGN with its clear sum of product decomposition makes its discontinuities explicit. Wang et al. make similar observations on a ReLU network by having a scalar bottleneck layer. Such modifications to practical architectures are necessary to even quantify anything about the internal structure of learned DNNs.

---

> ### Author Response · Authors · 2024-11-17
> **Response to reviewer n82d (Part 2 of 2)**
>
> 7. **Need for novel approaches in feature learning** : All current theoretical papers on feature learning are rather restrictive in their assumptions. They require orders of magnitude more overparameterization than what is typically present in practical neural nets, they require the use of a non-standard gradient descent training procedure or a non-standard setup like a bottleneck layer e.g. Theorem 1 in Wang et al.  Generalising these results to higher depth and getting meaningful insights into making training, inference or interpretation better for practice is a challenging task that has taken the community the better part of a decade with less than satisfactory results.
>
> 8. **DL theory has all its eggs in one basket** : Almost all current theoretical feature learning papers leverage compositionality of the label function as a key structural component in both results and proofs. It is quite natural to think that deep networks, which themselves derive power via composition of simple functions, also learn features via composition but this has yet to be concretely proven.
>
> 9. **Alternate possibility** : Our paper paper postulates an alternate possibility. The power of a deep network might be due to the fact that it learns separate features that interact (through a product in our case of DLGNs). This also gives a plausible mechanism of learning. The existence of multiple layers enables the ‘local’ nature of the discontinuity finding – each gating neuron operates in the context set by other layers, and hence, discontinuities that might be minor in a global context can become significant and drive the learning process. (Lines 61 to 65 in the paper). More concretely, in a data labeled by a COB-ODT, the root node halfspace would not be seen as a good direction for any gating hyperplane to move towards when seen globally. But when the data is restricted to random convex polyhedra (by say all gating neurons in a path except layer $\ell$) the gating function for the layer $\ell$ neuron in the path might see that the hyperplane corresponding to the root node is a good separator and move towards it. The fact that there are a huge number of paths means that this is likely to happen with at least a few paths.

---

> > ### Comment · Reviewer_n82d · 2024-11-21
> > **Response**
> >
> > The response resolves most of my concerns.
> > I will raise the score to 6.

---

### Official Review · Reviewer_D8rp · 2024-11-03

**Soundness:** 4
**Presentation:** 4
**Contribution:** 4
**Rating:** 8
**Confidence:** 3

**Summary:**

The authors introduce a model called the Deep Linearly Gated Network (DLGN) to study feature learning, specifically in binary classification tasks defined by an oblique decision tree labeling function. They use DLGN to test the hypothesis that during training, the model’s discontinuities move towards label function's discontinuities. The paper includes evaluations on dozens of open tabular datasets to compare DLGN with ReLU networks and tree-learning algorithms.

**Strengths:**

1. Clear writing. The authors study a specific class of problems with great clarity.
2. The authors' persistence in tackling a challenging yet manageable problem setting is commendable.
3. Like a black-box learner, the DLGN is able to learn non-linear features. Yet, it still provides mechanistic interpretability.
4. DLGN outperforms both tree-based and non-tree algorithms, as well as ReLU networks, in the oblique decision tree setting, while maintaining strong competitiveness on real-world tabular datasets.
5. The authors provide a framework that paves the way for future research and development.

**Weaknesses:**

I'm not seeing effective weaknesses.

**Questions:**

1. What's the purpose of defining the manifold $\mathcal{M}$ in line 133?
2. In line 239, Equation (4), is there a missing transpose on $\mathbf{u}_{i_1}^1$?
3. I find it difficult to understand how the computational cost of a forward pass for Equation (1) is less than twice that of a ReLU network with $mL$ nodes. Could the authors provide further clarification on this?

---

> ### Author Response · Authors · 2024-11-16
> **Response to reviewer D8rp**
>
> We thank the reviewer for the helpful feedback.
>
> 1. In a fuller version of the paper, we did have a paragraph connecting the manifold to DL architectures which we removed for brevity. The manifold still exists in the current version for reasons of generality. The definition of label discontinuties is applicable to any manifold while we study only hyperplanes. The DLGN model discontinuities are indeed just hyperplanes, but the geometry and parameterisation of the discontinuity manifold in a model is one of the core architecture design principles -- e.g. ReLU networks have "bent hyperplane" discontinuity manifolds, and CNNs have a more complex bouquet of bent hyperplanes instead of a single bent hyperplane due to the parameter tying and convolutional nature of the weights. We do not have any immediately insightful comments on why these structures should give better models on real data than simple hyperplanes like DLGNs.
>
> 2. $\mathbf{u}^1_{i_1}$ is a scalar.
>
> 3. The computational cost is approximately twice the cost of forward pass of a single ReLU network of the same size. This is because of the way $g_\pi$ is parameterised as a product of pairwise terms. A simple analogy is to see that a product of $L$ matrices of shape $m \times m$ can also be written as sum over all paths $\pi \in [m]^L$ but clearly matrix multiplication is not exponential complexity in $L$. See below for a more detailed explanation.
>
> 4. The model output can be simplified as $y(x) =. {\mathbf{u}^1}^\top D^1(x) U^2 D^2(x) \ldots U^L D^L(x) \mathbf{u}^{L+1}$ where $D^\ell(x)$ is a diagonal matrix corresponding to whether the nodes in the gating network in layer $\ell$ are active or not for input $x$. This is essentially just  multiplying $L$ matrices and hence has the same complexity as the forward pass of a ReLU network. We will add a short proof of the above statement in the appendix of the revised version.

---

> > ### Comment · Reviewer_D8rp · 2024-11-25
> >
> > Thank you. This response addresses my concerns. I look forward to the revised version.

---

### Official Review · Reviewer_tLyw · 2024-11-03

**Soundness:** 3
**Presentation:** 2
**Contribution:** 1
**Rating:** 5
**Confidence:** 4

**Summary:**

This paper proposes a novel model architecture (deep linearly gated network, DLGN) and studies the way in which this model seeks out “label discontinuities” in the data during training. This analysis is enabled the fact that we can enumerate such label discontinuities in a DLGN. The synthetic datasets with known discontinuities are generated by another model — an oblique decision tree (ODT). The authors also show how an ODT can be constructed from a trained DLGN, for the purpose of interpretability. Finally, the paper presents results of fitting a DLGN to several UCI regression tasks, comparing performance to several tree-based, kernel-based, and NN-based baselines.

**Strengths:**

- Originality. The DLGN is an interesting architecture that combines deep linear networks with the gating mechanism to construct a novel class of non-linear models. One could also treat DLGN as a novel decision tree parametrization which, when relaxed using a sigmoid, can be learned by back-propagation. To the best of my knowledge, DLGN is a novel, original model architecture, though its connection to soft decision trees should be studied more carefully.

**Weaknesses:**

- Lack of focus. At the moment, the paper’s focus is split between a study of feature learning using DLGN (chapter 5), and a study of the DLGN itself as an expressive, yet interpretable model architecture (chapter 6). To me these are two orthogonal contributions, and the paper would be stronger if authors focused on one of these.
- Significance. The significance of the proposed model and the feature learning study presented in the paper is not clear to me.
  1. Capturing “label discontinuities” (are these not simply decision boundaries?) is at the core of solving a classification problem, hence it is not surprising that a model which works well on the task has to discover such hyperplanes — I don’t see an alternative way that a model can solve a task. The real question is how an over-parametrized model can correctly identify high-dimensional decision boundaries given limited data — this is the main mystery in the theory of deep learning at the moment, and one that this paper doesn’t shed much light on.
  2. To understand the significance of the findings for deep learning, we would need to understand the relationship between a DLGN and a DNN. While it is nice (albeit, not surprising) to see how a DLGN uncovers the true “label discontinuities”, how can we know that a DNN will demonstrate the same behavior?
  3. While the proposed architecture is appealing due to potentially being both expressive and interpretable, results on the real (UCI) datasets suggests that DLGN is comparable in performance to standard tree algorithms. There is little evidence that we should prefer the proposed architecture to e.g. the well-studied random forests, which we could also argue to be “interpretable”. (I do not consider results on synthetic data to be good evidence, given the connection between ODTs used to generate the data and DLGN.)
  4. While the authors claim that the proposed architecture is interpretable, no interpretations of the models fit to real data are given. If interpretability if the main selling point of the architecture, I would expect a deeper analysis focused on interpretability.
- Presentation. Please consider moving the DLGN model diagram to the main text: it’s difficult to understand the model architecture from the formulas alone. Also, please try to stick to academic language, and avoid informal phrases like “well nigh impossible”, “handily outperform”, “succeed comfortably”, “about the same”, etc.

**Questions:**

- Authors suggest that they use a sigmoid instead of an indicator function to make the DLGN differentiable for training. Have authors considered using a temperature parameter for the sigmoid (potentially annealed during training), as common in other continuous relaxation methods?
- On lines 406-407 authors suggest that they use DBScan for clustering hyperplanes due to this algorithm being “robust to outliers” — why do authors expect outliers to be a significant issue here?
- How do authors explain the observation that certain layers of the DLGN are more prone to matching the true discontinuities than others  (Figure 2)?

---

> ### Author Response · Authors · 2024-11-15
> **First reply to reviewer tLyw  (Part 1 of 3)**
>
> Thank you very much for the detailed review. Responses to the review below. Some of these arguments are explicit in the paper while the rest are implicit. We will add extra remarks to make the implicit statement explicit in a revised version.
>
> 1. **DLGN as a tree reparameterization (point in the strengths section)** : This might be a misunderstanding, as the DLGN is NOT interchangeable with a decision tree. We do give a procedure for converting a trained DLGN into a decision tree based on the properties of its learning dynamics but an arbitrary DLGN cannot be converted into a decision tree. A core reason for the difference is that a data point can pass through any number of the m^L paths of a DLGN and the output is a sum of individual path values, as opposed to decision tree having a unique path for every input.
>
>
> 2. **Lack of focus**: Section 5 and 6 are closely related because the “clustering” of learned features in a trained DLGN (Section 5) is a crucial property  exploited in the decision tree construction (Section 6). Admittedly Section 5 could stand on its own and be expanded further without Section 6. But Section 6 plays an important role in making the DLGN a unique architecture. It demonstrates that the DLGN model allows for more control by the learner. For example, (say) a ReLU network model cannot be converted to any other known model type other than via painful retraining on surrogate data labeled by the ReLU network.
>
>
> 3. **Model finding label discontinuities is not surprising** : Broadly, yes, we agree. The title of the paper is perhaps a bit too simplistic. But the internals do more and give an alternative view of the term features itself and gives us the machinery and vocabulary to ask more meaningful questions. I am not sure the discontinuities can be called decision boundaries, as these have limited scope, e.g. consider the line corresponding to the root node 0 in Figure 1, the label function is only discontinuous along some sections of it. We have recently become aware of a concept known as the Gaussian surface area (from this year’s COLT best paper) which is very closely related to the notions that we have in mind and might play a crucial role in future work. Next two points are also related to this comment.
>
> 4. **Potential future lines of discussion being opened up**: The DLGN model is explicitly a weighted sum of product of halfspace indicator functions ($m^L$ in number) and hence we are somewhat justified in calling the individual halfspaces (only $mL$ in number) as features of the model. The halfspaces in the gating network seeking out the label function discontinuities (as demonstrated in Table 1) gives us a way to ask questions regarding parameter efficiency etc. Clearly, there is no need for multiple DLGN halfspaces to go towards the same discontinuity, but gradient descent (which is a greedy procedure) does it anyway. Maybe it would be possible to exploit this and have learning routines go beyond the greedy nature of gradient descent and build smaller models that perform similarly (say, identify two different gates going towards the same halfspace and pushing away one of them). This is however beyond our scope right now, but the very fact that such discussions are possible is the main point of this paper.
>
> 5. **What is the secret sauce in deep networks?**: Another aspect that the DLGN opens up is questioning the source of power in a deep network. It has been taken as an article of faith that this is due to composition of multiple simple layers. But this paper postulates that the power of a deep network might be due to the fact that it learns separate features that interact (through a product). This also sheds some  light on the mechanism by which training works in deep learning. The existence of multiple layers enables the ‘local’ nature of the discontinuity finding – each layer operates in the context set by other layers, and hence, discontinuities that might be minor in a global context can become significant and drive the learning process. (Lines  61 to 65 in the paper). More concretely, in a data labeled by a COB-ODT, the root node halfspace would not be seen as a good direction for any gating hyperplane to move towards when seen globally. But when the data is restricted to random convex polyhedra (by say all gating neurons in a path except layer $\ell$) the gating function for the layer $\ell$ neuron in the path might see that the hyperplane corresponding to the root node is a good separator and move towards it. The fact that there are a huge number of paths ($m^L$)  means that this is likely to happen with at least a few paths.

---

> ### Author Response · Authors · 2024-11-15
> **First reply to reviewer tLyw (Part 2 of 3)**
>
> 6. **Link between DLGN and DNN** : The tale behind the birth of the DLGN is rooted in the ReLU network. A ReLU network can be represented as $  y(x) = W_L* D_L* \ldots W_1* D_1 * W_0 * x $. Where the matrices $D_i$ are diagonal matrices that depend on $x$, taking values 1 or 0 depending on whether the corresponding neuron is active for the input $x$. When viewed as a function of $x$ the $i^{th}$ diagonal element of the matrix $D_l$ is rather complex, except for $l=1$ where it is simply equal to 1 on a halfspace and 0 outside it. DLGN simply makes the gating function for every neuron as a halfspace, whose parameters are given separately, i.e. in the above ReLU model output expression replace the diagonal matrices $D_\ell$ with diagonal matrices containing $\eta^\ell$ as defined in the paper.
>
> 7. **Can we say same statement about ReLU networks?**:Based on our current understanding of ReLU networks, it is not possible to show or check if a ReLU network discovers these discontinuities. (Where would we even look for these in the parameters of the DNN?). That we cannot demonstrate even such a non-surprising conclusion for the ReLU is exactly the reason we study the DLGN instead. In short, the significance of DLGNs is that there are no surprises and things are not “hidden”/inscrutable.
>
> 8. **DLGNs utility**: We don’t make the claim that DLGN is a better architecture that can be applied right now. The utility of the DLGN architecture is in being able to test our hypotheses regarding feature learning. Something that is not possible with deep linear networks (because they are essentially still linear) and ReLU networks (they are too complex). The experimental results are merely to establish that DLGNs outperform kernel methods, and hence learn non-trivial features. That they are competitive with random forests is merely an add-on bonus.
>
> 9. **DLGN interpretability**:  The source of complexity in ReLU nets lies in composition of nonlinear functions. Also, there is no meaningful notion of an “independent part” in a ReLU network – scalar weights connecting neurons are meaningless by themselves,  and neurons as real valued functions over the input space require almost as many parameters to describe as the entire network itself.  In the case of DLGN the source of power and complexity is just a sum of product of simple functions. While there are exponentially many ($m^L$) features, this complexity is due to a structured combination of merely $mL$ features. Also, each gating neuron is a meaningful part of the DLGN that can be compactly described as a halfspace in the input domain. This explicit sum of products nature of the DLGN makes it so that it can be almost considered a white box model.
>
> 10. **Tangential comment on the current state of interpretability** : We are not particularly enthusiastic about the current paradigms of interpretability (ala GradCAM, LIME, SHAP etc) which go to the extent of making the explanation itself as a prediction. The goodness of a prediction is judged by how it is found subjectively satisfactory rather than how faithful to the model the explanation is. We favor a more deconstructionist approach where we say “we understand something only if we can break it apart and put it together again”. The rise of *mechanisitc interpretability* (we are not fully sure of what an appropriate definition of this would be) in current literature suggests that current ML researchers are cognizant of this issue.

---

> ### Author Response · Authors · 2024-11-15
> **First reply to reviewer tLyw (Part 3 of 3)**
>
> Question 1: We indeed use a temperature hyperparameter, but freeze it to a constant during training. There is an interesting interplay between the temperature parameters and the initialization scale (i.e., for every temperature $\beta$ and init $w$, there exists an appropriate scalar $\gamma$ such that using temperature $1$ and init $\gamma w$ gives the same optimization trajectory.
>
> Question 2: We expect many of the trained DLGN gating hyerplanes to be useless/irrelevant to the label function, these hyperplanes need not be close to any of the label function discontinuities. These hyperplanes would be the outliers.
>
> Question 3: The first layer of a DLGN does not change much during the training. This is an artefact of the parameterization where the gating hyperplanes are represented as product of the parameters that the gradient descent operates on. If one maintains the effective linear transform for each layer directly as parameters, this artefact vanishes. (This would correspond to the variant DLGN-SF in the paper).

---

> ### Comment · Reviewer_tLyw · 2024-11-25
> **Follow-up on some of the authors' points**
>
> > DLGN is not a decision tree.
>
> I agree, but I would argue it is an alternative parameterization of a _soft_ decision tree ([Soft Decision Trees](http://www.cs.cornell.edu/~oirsoy/files/icpr21.pdf). _O. Irsoy, O. T. Yildiz, E. Alpaydin_. ICPR 21), which also computes a weighted combination of all paths in a tree.
>
> > Section 6 plays an important role in making the DLGN a unique architecture. It demonstrates that the DLGN model allows for more control by the learner. It demonstrates that the DLGN model allows for more control by the learner. For example, (say) a ReLU network model cannot be converted to any other known model type other than via painful retraining on surrogate data labeled by the ReLU network.
>
> I am not convinced demonstrating the ability to convert a DLGN to a decision tree helps the paper's goal of explaining feature learning in deep neural networks, given authors' claim that the "utility of the DLGN architecture is in being able to test our hypotheses regarding feature learning".
>
> > I am not sure the discontinuities can be called decision boundaries, as these have limited scope, e.g. consider the line corresponding to the root node 0 in Figure 1, the label function is only discontinuous along some sections of it.
>
> I believe this does not contradict the definition of a _non-linear_ decision boundary. And, again, the claim that for a model to do well on a classification task it has to capture decision boundaries is almost tautological.
>
> > But this paper postulates that the power of a deep network might be due to the fact that it learns separate features that interact (through a product).
>
> See the point below.
>
> > Based on our current understanding of ReLU networks, it is not possible to show or check if a ReLU network discovers these discontinuities. (Where would we even look for these in the parameters of the DNN?). That we cannot demonstrate even such a non-surprising conclusion for the ReLU is exactly the reason we study the DLGN instead. In short, the significance of DLGNs is that there are no surprises and things are not “hidden”/inscrutable.
>
> Continuing the point above, that's precisely what I see as a significant weakness of the current narrative. We see that this happens in a DLGN, but how can we be sure something like this happens in a ReLU network? Discussion on the connection between a DLGN and a ReLU network that authors provide in a separate paragraph is a start, and expanding on it, as well as showing how this connection suggests ReLU might demonstrate comparable behavior, would make the paper much stronger.
>
> I maintain my score for now.

---

> ### Author Response · Authors · 2024-11-27
> **Reply to follow-up**
>
> Thanks for the reply.
>
> We see the reviewers' concern about applicability of the message in the paper to practical neural net architectures like the ReLU network. Below, we make an attempt to address this concern.
>
> The DLGN is a surrogate architecture designed for a more detailed analysis of the feature learning mechanism in the learning of deep non-linear models by gradient descent. Thanks to the mathematical complexity of deep ReLU networks, all progress on our understanding of the learning mechanism is from surrogate architectures/algorithms which are not fully faithful to the original. An incomplete list follows:
>
> 1. Neural Tangent Kernel approaches: Theoretically, these require a large number of hidden neurons (a high degree polynomial in number of data points). It is not clear that any of the findings in this literature can be applied directly to practical ReLU nets, but they are generally acknowledged as giving a valid first-order understanding of the learning mechanism.
>
> 2. Deep linear networks: Claims to represent optimisation dynamics of deep networks, but is incapable of capturing feature learning.
>
> 3. Two layer architectures/Bottlenecks/Modified gradient descent: There is a host of literature on this line trying to explain the outperformance of GD on deep networks over kernel methods, but under a restricted two layer setting with a modified version of gradient descent.
>
> 4. Neural network gaussian processes: Does no gradient descent at all, and uses Bayesian conditioning instead.
>
> None of the above lines of work yield (currently) useful learning procedures that are actually deployed. None of the insights they generate are proven to apply to the more practically used deep architectures. Nonetheless, we value these results because it potentially enables the analysis of current algorithms or design of new and better learning algorithms.
>
> This brings us to a key point. The machine learning community is not really interested in unravelling the secrets of ReLU networks. It is more concerned with finding out the essence of what makes them work and hopes to find something better (in terms of train/test space/time complexity, more control and interpretability etc). This is the reason why alternate surrogate architectures are valuable. Each such architecture claims that the core reason for the success of deep networks is captured by them. Our work is no different in this sense. We propose the DLGN architecture motivated by the ReLU net (see point 6 in our response above) and claim that the core reason behind the success of deep networks (see point 5 in our response above) is captured by it.
>
> There are two core claims in the paper:
>
> 1. DLGN learns separate features that interact through a product. The existence of multiple layers enables the ‘local’ nature of the discontinuity finding – each layer operates in the context set by other layers.
>
> 2. ReLU networks and other deep architectures deployed in practice behave similarly.
>
> We provide some support for the first claim, and just appeal to the similarity of the DLGN to ReLU nets for the second.
>
> By the very nature of the goal for designing surrogate architectures, an appeal to similarity is all that is possible for the second claim (which is made implicitly or explicitly in all the architectures listed above).
>
> The value of our work lies in its ability to give a narrative to multi-layer learning beyond the usual 2 hidden layer theory. None of the other surrogate architectures mentioned above have such an ability. While admittedly we do not have an impressive looking theorem proving this narrative, it is possible to do so if enough extra assumptions are made (like for example optimising over $g_\pi$ directly, and using a block alternate minimization procedure where each weight vector is optimised while keeping others fixed) which would make the value of such a Theorem questionable.
>
> An example of the value of the narrative above is given below.  We could find sub-optimal behaviours of gradient descent, and potentially design novel algorithms beyond gradient descent. For example, consider the new Table 4 (which is in the same setting as Table 1, but with a wider DLGN architecture of $m=100$). While this architecture also attains a high train  accuracy, it performs worse than the $m=20$ architecture on the test data. It is easy to attribute this to “overfitting” and say that $m=20$ is the right hyperparameter choice. This is the only possible thing to do in an opaque architecture like the ReLU network. But in the DLGN model  with ODT label function setting, we can actually construct something like Table 4 and see that several of the internal node hyperplanes are not faithfully captured and the root node discontinuity is overrepresented. Thus we can give a deeper explanation of the failure than the surface level explanation of “overfitting”.
>
> We have added a few other deep learning phenomena candidate explanations to the Appendix.

---

### Author Response · Authors · 2024-11-27
**Summary of changes to the revised version**

We thank the reviewers for the detailed comments. We have made the changes asked for by the reviewers in the revised version. Here is a summary of the changes:

1. Put appendix and main paper in the same pdf.
2. Reduced some of the informal terminology in the main paper.
3. Fixed some errors in the figure in Appendix A.1 and made it more legible.
4. Added minor clarification to the local discontinuity coefficients and added a figure and discussion of the coefficient in Appendix A.4
5. Added proof for DLGN computation, i.e. equivalence of equation 1 and 5 in Appendix A.5
6. Added a short section in the appendix detailing the relation of DLGN to ReLU networks in Appendix A.6
7. Added a new section in the Appendix (A.7) discussing some deep learning phenomena and the candidate answers given by the feature learning narrative developed in the paper.

---

### Author Response · Authors · 2024-11-27
**Feature Learning Narrative Directions from DLGN. (Added in Appendix A7 of revision)**

We believe that the feature learning insights drawn from DLGN are applicable in general. More importantly, we believe that it can shed light on several phenomena that remain mysteries with deep networks. Some of the mysteries and candidate answers motivated from this paper are given below.

**Q1: Is neural network training using gradient descent parameter efficient?**

A1: No. The greedy nature of gradient descent, pushes multiple interconnected parts towards the same feature when it is potentially not required. e.g. In Table 4 the distances of learned DLGN hyperplanes (with width m = 100) to the ODT hyperplanes are given. While the root node hyperplane is discovered separately by multiple gating hyperplanes, some of the other internal nodes are left in the lurch. This causes poor generalisation. Potentially, this narrative can be exploited to make gradient descent less greedy.

**Q2: How does gradient descent discover the label function discontinuities in high dimensional input space?**

A2: The full labelling function may be far from being a linear separator, but when data is restricted within a small enough scope, one of the manifolds making up the decision boundary could very well be a good classifier (this argument uses the insight that even in high dimensional data, if it is linearly separable, picking up the separator can be done in a compute and data efficient manner). This causes that manifold to be picked up by one of the components of the deep network, and thereby making the rest of the learning problem easier and kickstarting a virtuous cycle.

**Q3: Why does neural network pruning enable better learning of smaller architectures than training the smaller architecture from scratch?**

A3: We assume effective learning happens only in problems where the number of label function discontinuities is small. Significant parts of the neural network are simply not necessary to represent the discontinuities, which can be done by a small fraction of the trained deep network model. A large width network is still better for training, because (say) doubling the number of neurons per layer increases the number of paths by a factor of $2^L$. This increases the chances of some path picking a right scope during training enabling the learning of the appropriate label function discontinuity. Once the learning is complete, a significant fraction of the network which were unlucky to not have gotten a good scope can be removed.



**Q4: What is the role of layers in deep networks? Is increasing the number of layers always beneficial?**

A4: The main role of layers is to give context/scope to other layers. For example, with a depth 4 ODT labelling function, a DLGN with 3 or lesser number of hidden layers would not be able to give a good scope to any neuron. A DLGN with 5 or more layers is just unnecessary. This is reflected in our experiments as well, where depth 4 DLGN performed best for a depth 4 ODT labelling function.

---

### Meta-Review · Area_Chair_b1k1 · 2024-12-21

**Metareview:**

The paper proposes the deep linearly gated network to analyze feature learning in deep neural networks. It highlights how label function discontinuities guide model function discontinuities during training, using a novel architecture for interpretable results. Contributions include mechanistic insights into feature learning, a direct link to decision tree extraction, and competitive performance on synthetic and real-world datasets.

The novelty of the concept proposed by this paper is highly appreciated. The clear presentation also adds to its contribution. Further extensions and more rigorous experimental validation are noted, but the paper is judged to be in satisfactory form at this time.

**Additional Comments On Reviewer Discussion:**

tLyw acknowledged its novelty but noted the importance of some concepts; D8rp appreciated its contribution and clarity; n82d acknowledged its contribution but raised concerns about extensibility, etc.; xQ4v appreciated the interesting nature of the study but noted the limitations of the experiment. In any case, the overall evaluation was high.

---

### Decision · Program_Chairs · 2025-01-22

Accept (Poster)